# CaMKII oxidation is a critical performance/disease trade-off acquired at the dawn of vertebrate evolution

Qinchuan Wang [1✉], Erick O. Hernández-Ochoa[2], Meera C. Viswanathan[1], Ian D. Blum [3], Danh C. Do[1], Jonathan M. Granger[1], Kevin R. Murphy [1], An-Chi Wei[4], Susan Aja[5,6], Naili Liu[7], Corina M. Antonescu[8], Liliana D. Florea [8], C. Conover Talbot Jr. [9], David Mohr[10], Kathryn R. Wagner[3,7], Sergi Regot [11], Richard M. Lovering[12], Peisong Gao [1], Mario A. Bianchet [3,13], Mark N. Wu[3], Anthony Cammarato[1], Martin F. Schneider[2], Gabriel S. Bever[1,14] & Mark E. Anderson [1✉]

Antagonistic pleiotropy is a foundational theory that predicts aging-related diseases are the result of evolved genetic traits conferring advantages early in life. Here we examine CaMKII, a pluripotent signaling molecule that contributes to common aging-related diseases, and find that its activation by reactive oxygen species (ROS) was acquired more than half-a-billion years ago along the vertebrate stem lineage. Functional experiments using genetically engineered mice and flies reveal ancestral vertebrates were poised to benefit from the union of ROS and CaMKII, which conferred physiological advantage by allowing ROS to increase intracellular $Ca^{2+}$ and activate transcriptional programs important for exercise and immunity. Enhanced sensitivity to the adverse effects of ROS in diseases and aging is thus a trade-off for positive traits that facilitated the early and continued evolutionary success of vertebrates.

[1] Department of Medicine, Johns Hopkins School of Medicine, Baltimore, MD, USA. [2] Department of Biochemistry and Molecular Biology, University of Maryland School of Medicine, Baltimore, MD, USA. [3] Department of Neurology, Johns Hopkins School of Medicine, Baltimore, MD, USA. [4] Department of Electrical Engineering, Graduate Institute of Biomedical Electronics and Bioinformatics, National Taiwan University, Taipei, Taiwan. [5] Department of Neuroscience, Johns Hopkins School of Medicine, Baltimore, MD, USA. [6] Center for Metabolism and Obesity Research, Johns Hopkins School of Medicine, Baltimore, MD, USA. [7] Center for Genetic Muscle Disorders, Kennedy Krieger Institute, Baltimore, MD, USA. [8] Johns Hopkins Computational Biology Consulting Core, Baltimore, MD, USA. [9] Institute for Basic Biomedical Sciences, Johns Hopkins School of Medicine, Baltimore, MD, USA. [10] Johns Hopkins School of Medicine Genetic Resources Core Facility, Baltimore, MD, USA. [11] Department of Molecular Biology & Genetics, Johns Hopkins School of Medicine, Baltimore, MD, USA. [12] Department of Orthopaedics, University of Maryland School of Medicine, Baltimore, MD, USA. [13] Department of Biophysics and Biophysical Chemistry, Johns Hopkins University, Baltimore, MD, USA. [14] Center for Functional Anatomy & Evolution, Johns Hopkins School of Medicine, Baltimore, MD, USA. ✉email: qinchuan.wang@jhmi.edu; mark.anderson@jhmi.edu

Governments, insurers, and individuals devote enormous resources to mitigate the risks and consequences of cardiovascular disease and cancer, which are the major non-communicable causes of death in modern society[1]. The most important risk factor for these diseases, however, is aging-related deterioration[2], and the leading evolutionary theory explaining such aging-related disease is antagonistic pleiotropy[3]. In general, the power of natural selection to either promote beneficial or remove detrimental effects from a population/lineage is thought to wane as the realization of those effects moves increasingly late in life. Evolution within this "selection shadow" is predicted to take on a more stochastic pattern with the unfortunate consequence that deleterious, aging-related mutations can accumulate with relative freedom[4]. A potential shortcoming of this mutation accumulation model is that the attenuation of selection's negative influence fails to fully explain the seemingly entrenched nature of many aging-related diseases because it leaves intact the probability that drift can also remove these detrimental traits (especially when combined with even weak coefficients of negative selection). Antagonistic pleiotropy helps complete this theoretical structure by predicting that the genetic traits responsible for some or all aging-related diseases—and even aging itself—are maintained because at some level they are being actively promoted by positive selection. What then is the target of this positive selection? It seems unlikely that the selective advantage is to be found in the late-stage problems these traits incur. A more reasonable prediction is that the traits convey benefits to earlier stages of life when selection is more efficacious. Despite the firm theoretical foundation of antagonistic pleiotropy, the theory has long been hampered by a relatively low level of empirical support. This disparity perhaps reflects the tendency of natural selection to quickly eliminate the polymorphisms required to identify and test candidate genes[5]. And while established support certainly is growing[6], when one considers the number of potentially relevant pathways and both the biological and societal implications of the theory, it is clear that more work is needed.

Reactive oxygen species (ROS) contribute to aging-related diseases by damaging cellular components and dysregulating ROS-sensitive proteins[7,8]. The fact that ROS contribute to aging and aging-related diseases, but are also capable of beneficial roles[9] suggests that proteins responsible for ROS sensing might be part of an antagonistic pleiotropic genetic program that drives aging-related diseases, including cardiovascular disease and cancer[10]. The $Ca^{2+}$-and calmodulin-dependent protein kinase II (CaMKII) is a prominent example of a ROS-sensitive signaling protein[11]. Oxidative activation of CaMKII is enabled by oxidation at cysteine 281 and methionine 282 (CM; CaMKIIα) or methionine 281 and 282 (MM; CaMKIIγ, δ and β) in the regulatory domain after the enzyme is initially activated by elevated $Ca^{2+}$-bound calmodulin (Fig. 1a)[11]. CaMKII activity using the oxidative pathway (ox-CaMKII) is elevated in tissues of patients with cardiovascular diseases[12,13], human cancer cell lines[14], and asthmatic human pulmonary epithelium[15]. We have reported that mutant mice where CaMKIIδ was rendered insensitive to ROS, by knock-in replacement of methionines 281 and 282 with valines, were protected against a range of diseases that involve elevated oxidative stress, including cardiac arrhythmia, ischemia-reperfusion injury, sudden death, and asthma[12,13,16–18]. Given that the detrimental roles of ox-CaMKII are apparently tolerated by natural selection in humans and mice, we hypothesized that ox-CaMKII is a critically important, but heretofore unrecognized, molecular example of antagonistic pleiotropy, and thus predicted that ox-CaMKII would contribute to beneficial roles favorable to natural selection.

Here, we show that ox-CaMKII first evolved in the ancestral vertebrates and is highly conserved in vertebrate lineages.

Removing ox-CaMKII in mice suppresses $Ca^{2+}$-mediated signaling in skeletal muscles, leading to reduced exercise performance and dampened transcriptional response to exercise. The ox-CaMKII-deficient mice also show a reduced transcriptional response in mast cells after degranulation. Conversely, introducing ox-CaMKII to flies by CRISPR-mediated gene editing enhances their motor performance and cardiac contractility in a ROS-dependent fashion. However, gaining ox-CaMKII also sensitizes the flies to pathological ROS and shortens their lifespan. Our results suggest that ox-CaMKII is a performance/disease trade-off consistent with the precepts of antagonistic pleiotropy.

## Results

**MM/CM evolved as a concise ROS sensor in ancestral vertebrates.** The evolutionary fixation of ox-CaMKII in humans and mice serves to obscure the processes responsible for that fixation, which can include drift, selective pressure, or some combination of the two. Phylogenetic variation, however, can help to reveal this history. Therefore, we surveyed all reported sequences of CaMKII for the presence or absence of the MM/CM module in the regulatory domain that is necessary for ox-CaMKII. The MM/CM module is extremely conserved among crown vertebrates, with 97.94% of vertebrate CaMKII possessing the CM/MM module (Supplementary Fig. 1 and Supplementary Table 1), and 100% of vertebrate species having at least one copy of CaMKII that has the MM/CM module (Supplementary Table 1). In contrast, only 2.23% of invertebrate CaMKII have the MM/CM module (Supplementary Fig. 1 and Supplementary Table 1). The phylogenetic distribution of the MM/CM module supports the inference that an oxidation-sensitive amino acid pair in the regulatory domain is a derived feature of vertebrates, one that emerged on the vertebrate stem lineage ~650 million years ago (Fig. 1b). The emergence of the MM/CM module in the vertebrate stem lineage is unique amongst regulatory domain residues of CaMKII capable of conferring CaMKII activity by post-translational modifications (see Fig. 1a). The other notable residues in the vicinity of MM/CM are threonine 287 (T287, numbered in accordance with CaMKIIγ isoform) for phosphorylation[19,20], and serine 280 (S280) for O-GlcNAcylation[21]. Predating the MM/CM by about 400 million years, T287 appeared on the same branch as CaMKII itself, while the S280 pathway evolved slightly later (but still prior to the origin of Metazoa), and both residues are conserved since their first appearance (Fig. 1b). This clear phylogenetic demarcation between the vertebrate and invertebrate lineages by the MM/CM module provides us with a pivotal opportunity to experimentally test the predictions of antagonistic pleiotropy from the perspective of ox-CaMKII.

To ascertain that gaining MM/CM at positions 281 and 282 was sufficient to convert ancestral CaMKII into a ROS sensor[22], we substituted the ROS-resistant VV residues in the sole copy of CaMKII in *Drosophila melanogaster* with MM (Supplementary Fig. 2). Since fly CaMKII structure has not been reported, we used comparative modeling based on the structure of an inactive human CaMKIIα to evaluate the impact of this substitution on fly CaMKII (Supplementary Fig. 3). The model showed that the VV-containing region of fly CaMKII shares a high structural similarity with human CaMKIIα, and similar to the ROS-sensing CM residues in human CaMKIIα, the VV residues in fly CaMKII are exposed to solvent (Supplementary Fig 3a). Substituting the VV residues by MM in fly CaMKII can be accommodated without structural perturbation; thus, the MM residues in the humanized fly CaMKII are also exposed to the solvent where they can be attacked by ROS (Supplementary Fig. 3a, b). Furthermore, hydrophobic cluster analysis[23] showed

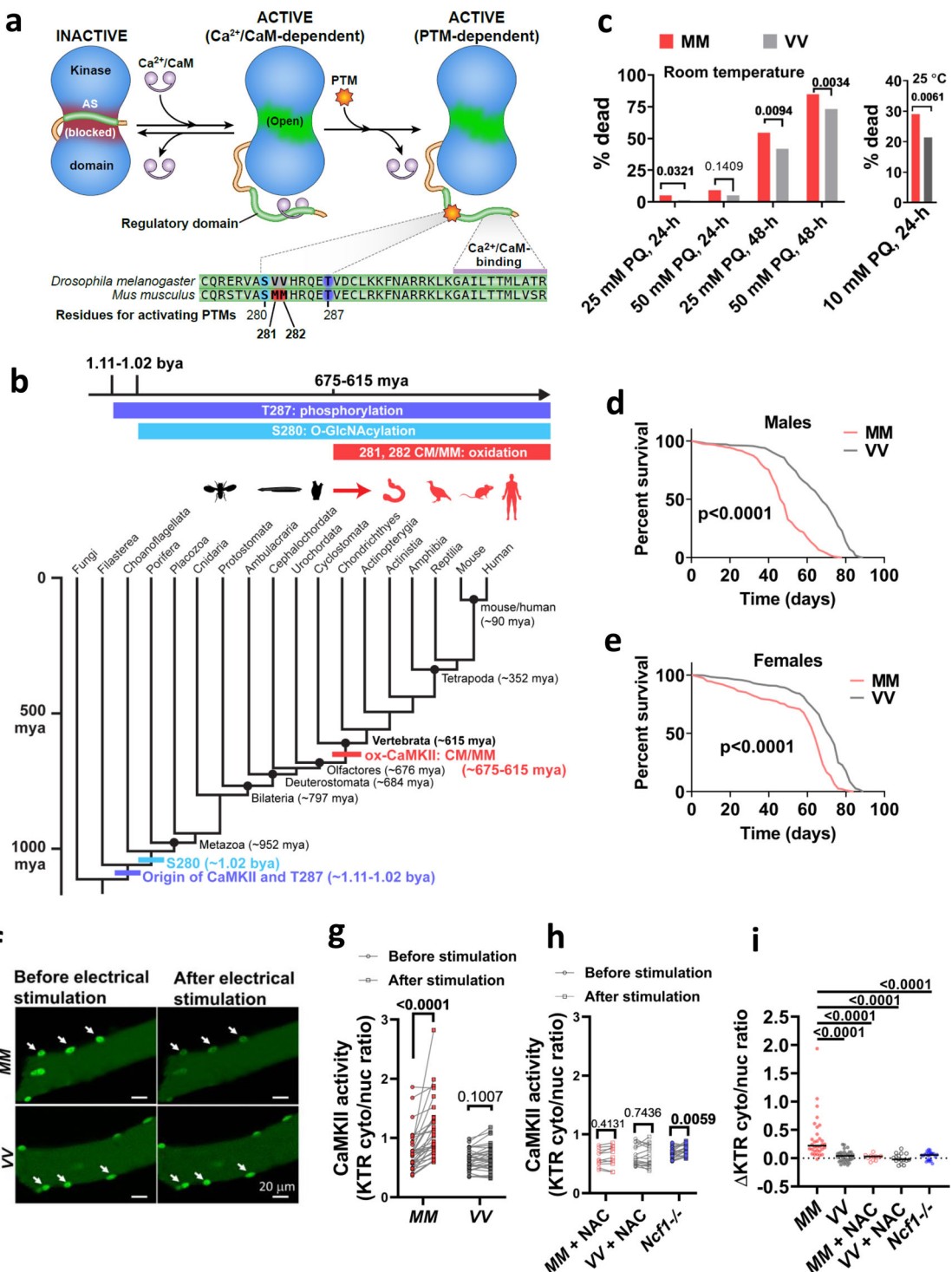

that replacing VV with MM conserves the characteristics of the local environment (Supplementary Fig. 3c). Thus fly CaMKII shares a similar structure to human CaMKIIα, and substituting the VV with MM residues does not alter the structure of the CaMKII regulatory or catalytic domains (Supplementary Fig. 3d). Our model predicts that replacing the VV with MM residues in fly CaMKII should preserve the functions of CaMKII while conferring it with ROS-sensing ability. Consistent with the prediction that the MM residues do not render fly CaMKII inactive, the *CaMKII^MM^/CaMKII^MM^* flies (referred to as MM flies hereafter) have similar fecundity (Supplementary Fig. 4a, b) as wild-type (WT) flies (referred to as VV flies), and are grossly

indistinguishable from VV flies. Furthermore, the MM flies showed significantly higher mortality when fed sucrose solutions dosed with paraquat (Fig. 1c), a ROS-inducing toxin[24], confirming a functional consequence to MM oxidation. The increased mortality of MM flies under paraquat treatment was not due to exaggerated oxidative stress compared to VV flies, because ROS levels indexed by ROS-liable aconitase activity[25] showed no difference between MM and VV flies either before or after paraquat treatment (Supplementary Fig. 5). Therefore, replacing the VV with MM residues in fly CaMKII converts CaMKII into a ROS sensor that sensitizes flies to paraquat. Because paraquat induces a supraphysiological level of ROS, we further evaluated

**Fig. 1 The MM motif in CaMKII arose in vertebrates, allowing CaMKII activation by ROS. a** CaMKII activation is initiated by binding $Ca^{2+}$/CaM, but $Ca^{2+}$/CaM-independent CaMKII activity is sustained by oxidation (MM 281/282) and other post-translational modifications of the regulatory domain (S280 and T287). The CaMKII regulatory domain sequences of *Drosophila melanogaster* CaMKII and *Mus musculus* CaMKIIγ are shown. **b** Phylogenetic origin and conservation of key residues for the activating PTMs of CaMKII, divergence time estimates from[77]. mya million years ago, bya billion years ago. **c** Replacing the VV residues of *Drosophila melanogaster* CaMKII with MM increased mortality caused by exposure to 25 mM and 50 mM paraquat (PQ) incorporated in 5% sucrose solution at room temperature (21 °C, left panel) after 24 h (24 h) and 48 h (48 h), and by exposure to 10 mM paraquat at 25 °C after 24 h (right panel). *P* values from Fisher's exact test, two-tailed, are shown above the square brackets. $N = 216$ and 217 respectively for MM and VV flies treated at room temperature; $n = 510$ for both genotypes of flies treated by either control or paraquat solutions under the 25 °C test condition. **d** Survival curves of MM ($n = 239$) and VV ($n = 299$) males and **e** MM ($n = 270$) and VV ($n = 300$) females at 25 °C, log-rank test, two-tailed. **f** Representative confocal micrographs of MM and VV mouse FDB muscle fibers expressing CaMKII-KTR before and after field electrical stimulation. Arrows indicate nuclei. **g** Quantification of CaMKII activity (cytosolic to nuclear CaMKII-KTR signal ratio) in MM and VV fibers before and after electrical field stimulation (paired, nonparametric Wilcoxon tests, two-tailed, $n = 32$ nuclei from 12 MM fibers and $n = 33$ nuclei from 11 VV fibers). **h** Quantification of CaMKII activity before and after electrical field stimulation in the presence of 2 mM NAC or *Ncf1* deletion (paired, nonparametric Wilcoxon tests, two-tailed. $n = 11$ nuclei from 5 MM fibers, $n = 16$ nuclei from 6 VV fibers, $n = 24$ nuclei from 9 *Ncf1-/-* fibers. **i** changes of CaMKII-KTR cytosolic to nuclear ratio due to electrical stimulation of individual nuclei from panel **g** and **h**, horizontal lines indicate median, Kruskal–Wallis test followed by Dunn's nonparametric multiple comparisons test, two-tailed. Source data are provided in the Source Data file.

the consequence of CaMKII-mediated ROS sensing in the context of endogenous ROS. Endogenous ROS increases with aging and adversely contributes to aging-associated mortality through ROS-sensitive signaling pathways[9]. Strikingly, MM flies exhibited significantly shortened lifespan at both 25 °C (Fig. 1d, e) and 29 °C (Supplementary Fig. 6a, b), suggesting that the CaMKII oxidation in MM flies happens in the presence of endogenous ROS, and contributes to aging-associated mortality. We noted that male flies are more susceptible than females to the presence of the MM module at both 25 °C (Fig. 1d, e) and 29 °C (Supplementary Fig. 6a, b). Furthermore, under the condition of heat stress at 29 °C, the detrimental effects of the MM module further diminish in females. This sex difference may reflect the difference in ROS scavenging capacities between sexes. Specifically, females have a better ROS scavenging capacity during aging[26]. Thus, females may have less ROS to hyperactivate MM-CaMKII during aging and thus tolerate the MM module better. We speculate that because MM-CaMKII contributes less to aging in females, its detrimental effect is largely masked in females at 29 °C, where heat stress may accelerate aging through mechanisms in addition to oxidative stress.

Taken together, inserting the MM module in flies confers elevated and detrimental sensitivity to ROS in these invertebrates, which may be the reason that ox-CaMKII is underrepresented among invertebrate lineages as a whole. That MM-dependent sensitivity to ROS is shared between mouse and *Drosophila melanogaster* supports the inference that this sensitivity is homologous between the two species and was ultimately inherited from their most recent common ancestor—the ancestral crown bilaterian. Specifically, this evolutionary pattern represents a strong level I phylogenetic inference because the feature is shared by both sides of the extant phylogenetic bracket with no evidence from intervening lineages that would negate or significantly complicate the null hypothesis of homology[22,27]. It thus follows that ox-CaMKII-mediated detrimental sensitivity to ROS was in place when the oxidative pathway first evolved along the vertebrate stem lineage with its subsequent conservation among vertebrates serving as strong evidence that these sequences are not evolving randomly but are maintained by positive selection. Given that this conserved pattern extends across the entire breadth of the vertebrate crown clade, whatever fitness benefits are responsible for its underlying positive selection must have been in place before the origin of this crown and thus present along some length of the vertebrate stem lineage. It follows that these benefits are likely related to an emerging evolutionary scenario that set these vertebrate ancestors apart from their invertebrate relatives.

**Ox-CaMKII promotes exercise performance in mice by enhancing skeletal muscle function.** The vertebrate stem lineage was witness to a major shift in behavioral ecology (from sessile filter feeders to active predators) that set the stage for the modern (crown) vertebrate radiation and eventually our own evolutionary origin[28]. At some level, all of these innovations promote an increasing level of activity that must be enacted through the skeletal (striated) musculature. CaMKII activity contributes to skeletal muscle function[29,30], so we hypothesized that gaining the MM module allowed ROS to enhance skeletal muscle performance through ox-CaMKII. To determine the role of MM residues for ROS-induced CaMKII activity in muscle fibers, we developed a knock-in mouse where the MM residues of CaMKIIγ were replaced with VV (Supplementary Fig. 7a, b, homozygous $CaMKII\gamma^{VV}/CaMKII\gamma^{VV}$ mice are referred to as VV mice hereafter). We selected *Camk2g* as the background because we found that CaMKIIγ is the most abundant isoform in mouse skeletal muscle (Supplementary Fig. 8a). The knock-in mutation did not change the relative expression levels among the four CaMKII isoforms in the muscles, indicating that the predominant CaMKII isoform in VV muscles is the ROS-resistant, VV-containing CaMKIIγ (Supplementary Fig. 8b). To measure dynamic changes in CaMKII activity in muscle fibers, we developed a fluorescent reporter that translocates from the nucleus to the cytosol in response to increased activity of CaMKII (kinase translocation reporter, or KTR, Supplementary Fig. 9a–e and Methods)[31]. We introduced the CaMKII-KTR into the flexor digitorum brevis (FDB) skeletal muscles of MM (referring to WT mice) and VV mice by electroporating plasmids encoding the reporter (see "Methods")[32]. In isolated FDB muscle fibers expressing the KTR, we found that the increase of CaMKII activity in response to electrical stimulation (see Fig. 2g for stimulation protocol) required the MM motif (Fig. 1f–i). Treatment with the antioxidant N-acetylcysteine (NAC) eliminated the rise in CaMKII activity in MM (WT) FDB fibers in response to stimulation but had no further effect in VV fibers (Supplementary Fig. 10 and Fig. 1h–i). We next used a genetic approach to reduce ROS by isolating muscle fibers from $Ncf1^{-/-}$ mice that lack p47[33], an essential protein cofactor for NADPH oxidases that are an important source of ROS in skeletal muscles[34–36]. Although CaMKII activity still increased in $Ncf1^{-/-}$ fibers after stimulation (Fig. 1h), the magnitude of the increase was significantly smaller than that in the MM fibers (Supplementary Fig. 10 and Fig. 1i). The results showed that ROS contribute to the activation of myofiber CaMKII, a process dependent on the MM module of CaMKII.

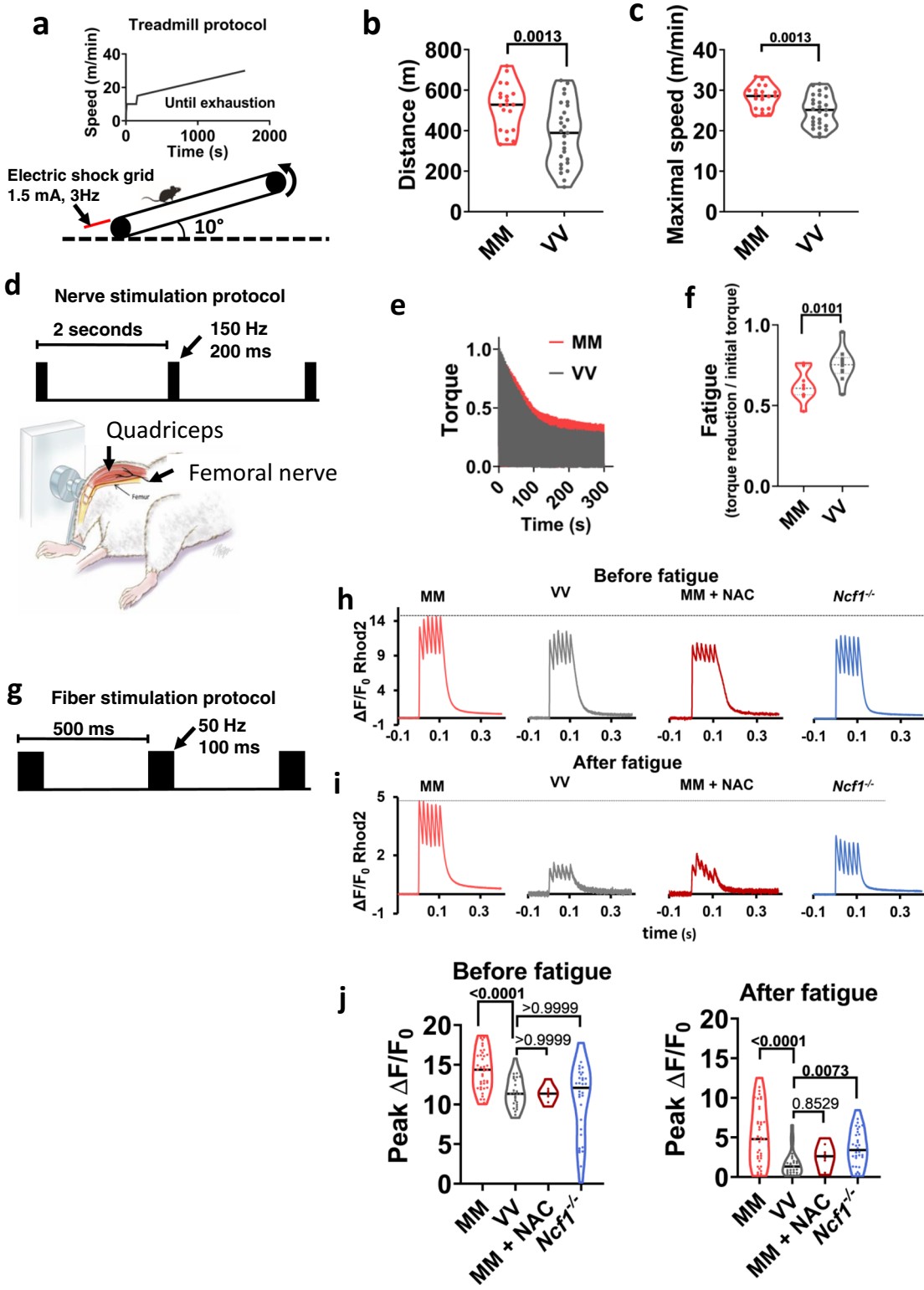

To test for ox-CaMKII enhanced skeletal muscle performance, we subjected MM (WT) and VV mice to maximal coerced treadmill exercise (Fig. 2a). MM (WT) mice ran farther (Fig. 2b) and attained higher maximal speeds (Fig. 2c) compared to VV littermates. Although oxidative stress and the MM motif were necessary for normal CaMKII activity in stimulated muscle fibers (Fig. 1f–i and Supplementary Fig. 10), the reduced exercise performance in VV mice could be due to factors extrinsic to the muscles, such as metabolism or motivation. We found no

significant differences in the body weight, lean mass, or fat mass between MM (WT) and VV mice (Supplementary Fig. 11a–c). Nor did we detect differences in blood lactate concentration at rest or after running (Supplementary Fig. 12a), blood glucose before or immediately after running (Supplementary Fig. 12b), or significant differences in glucose tolerance or insulin sensitivity in sedentary MM and VV mice (Supplementary Fig. 12c, d). Finally, MM and VV mice engaged in similar amounts of voluntary wheel running (Supplementary Fig. 12e), and exhibited similar oxygen

**Fig. 2 Ox-CaMKII supports exercise performance and enhances $Ca^{2+}$ transients in mouse skeletal muscle fibers. a** Protocol for treadmill exercise. **b** Running distance and **c** maximal speed attained prior to exhaustion. Unpaired Welch's $t$ test, two-tailed, $P$ values shown above the square brackets, $n = 20$ MM (WT) mice and $n = 27$ VV mice. **d** Experimental apparatus and electrical stimulation protocol for assessing quadriceps muscle performance in vivo. **e** Averaged traces of quadriceps torque (normalized against maximum) from MM and VV mice during optimized nerve stimulation, $n = 10$ VV mice, and $n = 11$ MM mice. **f** Quantification of fatigue defined by torque reduction divided by initial torque of each individual mouse, as shown in **e**. Unpaired Welch's $t$ test, two-tailed, $P$ value shown above the square bracket. **g** Protocol for field electrical stimulation of isolated FDB muscle fibers loaded with the $Ca^{2+}$-sensitive fluorescent dye Rhod2. Rhod2 fluorescence during one cycle of electrical stimulation in MM and VV fibers, an MM fiber treated by NAC, and a $Ncf1^{-/-}$ fiber, before **h** and after **i** fatigue. **j** Quantification of peak $Ca^{2+}$ transients as measured in **h** and **i**. Both before and after fatigue: $n = 42$ MM, $n = 27$ VV, $n = 8$ MM + NAC, and $n = 32$ $Ncf1^{-/-}$ fibers; after fatigue: $n = 40$ MM, $n = 27$ VV, $n = 8$ MM + NAC, and $n = 32$ $Ncf1^{-/-}$ fibers; $P$ values are shown above the brackets; Kruskal–Wallis test followed by Dunn's nonparametric multiple comparisons tests, two-tailed, comparing all other groups to VV fibers in **j**. Horizontal lines in **b**, **c**, **f**, and **j** indicate medians. Source data are provided in the Source Data file.

consumption rates, respiratory exchange ratios and energy expenditures while accessing the running wheels (Supplementary Fig. 12f–h). These negative findings suggest that the reduction in forced running performance in VV mice was not due to muscle extrinsic factors including reduced motivation for running or changed metabolism.

To assess the potential consequences of ox-CaMKII on muscle function, we measured the reduction in quadriceps contractility in response to repeated maximal isometric contractions elicited by repetitive electrical stimulation of femoral nerves, a validated test of muscle fatigue (Fig. 2d) that allows for the direct examination of muscle function in vivo[37]. The VV mice exhibited earlier and enhanced fatigue compared to MM (WT) littermate mice (Fig. 2e, f). The reduced performance of the VV mice was unlikely due to developmental defects or gross pathological remodeling, as we found that the VV and MM (WT) mice have similar muscle weight to body weight ratios and grip strength (Supplementary Fig. 13a–g). In addition, the contents of mitochondrial complexes in muscles (Supplementary Fig. 14a), and oxidative phosphorylation and glycolysis capacities of isolated FDB muscle fibers were similar between MM (WT) and VV mice (Supplementary Fig. 14b, c). Furthermore, we found no significant change in the fatigue-resistant type I fibers and noted significant but subtle switches among type II fibers in VV quadriceps muscles, which were unlikely to explain the reduced endurance capacity (Supplementary Fig. 15a, b). Although these results do not rule out potential roles of ox-CaMKII extrinsic to muscle, they support a view that ox-CaMKII enhances dynamic responses to exercise and skeletal muscle performance, potentially by a muscle intrinsic factor(s).

Intracellular $Ca^{2+}$ grades myofilament interactions, and fatigue is marked by reduced intracellular $Ca^{2+}$ transients[38]. We used a validated in vitro model of skeletal muscle fiber fatigue[39], under conditions where we monitored the intracellular $Ca^{2+}$ transients (see "Methods" and Fig. 2g). The VV fibers had reduced $Ca^{2+}$ transients under basal (Fig. 2h, j) and fatigued (Fig. 2i, j) conditions compared to MM (WT) counterparts. To test whether the exaggerated fatigue $Ca^{2+}$ phenotype in VV muscle fibers was a consequence of ROS signaling, we treated MM (WT) fibers with the antioxidant NAC (N-acetylcysteine). The MM fibers exposed to NAC phenocopied the $Ca^{2+}$ release profiles measured in VV fibers (Fig. 2h–j). Similar to NAC-treated MM (WT) fibers, the $Ncf1^{-/-}$ fibers shared a phenotype of diminished $Ca^{2+}$ transients resembling VV muscle fibers (Fig. 2h–j). Taken together, we interpret these data as supporting a model where ox-CaMKII contributes to enhanced skeletal muscle performance, at least in part, by connecting ROS to mobilization of intracellular $Ca^{2+}$.

**Ox-CaMKII coordinates transcriptional responses to exercise, ROS, and immune activation.** Exercise imposes metabolic, mechanical, and redox stress on skeletal muscle leading to transcriptional adaptation that is partly orchestrated by CaMKII[40,41].

We measured transcriptional responses to submaximal exercise in skeletal muscles, comparing poly(A)$^+$ transcriptomes by RNA sequencing from MM (WT) and VV littermate mice under identical conditions of speed, time, distance, and feeding conditions (Fig. 3a and see "Methods"). Principal components analysis showed that sedentary MM (WT) and VV muscles had very similar transcriptional profiles (Fig. 3b). In contrast, transcriptional responses to exercise by VV muscles were present but diminished compared to MM (WT) (Fig. 3b). We found that 582 genes were significantly up- or downregulated (multiple-test false discovery rate-adjusted $q$-value < 0.05) in the MM (WT) samples, whereas only 216 genes reached the same threshold of $q < 0.05$ in the VV muscles (Fig. 3c and Supplementary Table 2). Among the significantly changed genes in VV muscles, most (180 or 83%) were recapitulated by the MM (WT) muscles. To further compare the transcriptional responses of MM (WT) and VV muscles at the level of individual genes, we ranked the exercise-responsive genes identified in the MM (WT) muscles based on their $\log_2$ (fold change) values, as diagrammed (Fig. 3d, left panel), and plotted these genes in heat map palettes (Fig. 3d, middle panel). The changes of the same set of genes in the VV muscles were shown in separate palettes (Fig. 3d, right panel), but in the same order as that of the MM palettes. The results show that, in general, the MM module augmented the responses of individual genes to exercise. While most exercise-responsive genes preserved their qualitative responses in VV muscles, these responses are quantitatively and heterogeneously depressed. We then plotted the genes whose changes due to exercise reached statistical significance only in the VV muscles (Fig. 3e). Again, we found that the changes of most of these genes were qualitatively similar in the MM muscles. The results suggest that ox-CaMKII plays important roles in the acute transcriptional response of skeletal muscles to exercise. The global dampening of transcriptional response in VV muscles suggests that ox-CaMKII regulates exercise-responsive gene expression largely by augmenting the $Ca^{2+}$ signals in muscles during exercise (Fig. 1h–j), which acts as an upstream second messenger for the global transcriptional response through a myriad of $Ca^{2+}$ sensitive signaling pathways[41]. On the other hand, the heterogeneous effects from loss of ox-CaMKII on different genes (Fig. 3d, e) suggest that some genes may be more sensitive to the disturbance in $Ca^{2+}$ signaling, or that they are direct downstream targets of ox-CaMKII.

We next used QIAGEN ingenuity pathway analysis[42] to extract biological pathway information from genes that showed a large shift (more than $\pm 2\sigma$) in expression in response to exercise. We found that in the MM (WT) muscles responding to exercise, eight out of the ten most significantly enriched biological function terms were related to inflammation (Fig. 3f). Strikingly, exercise-induced lesser changes of the same set of inflammatory responses in the VV muscles (Fig. 3f). Conversely, in the VV muscles, the top ten most enriched biological functions responding to exercise

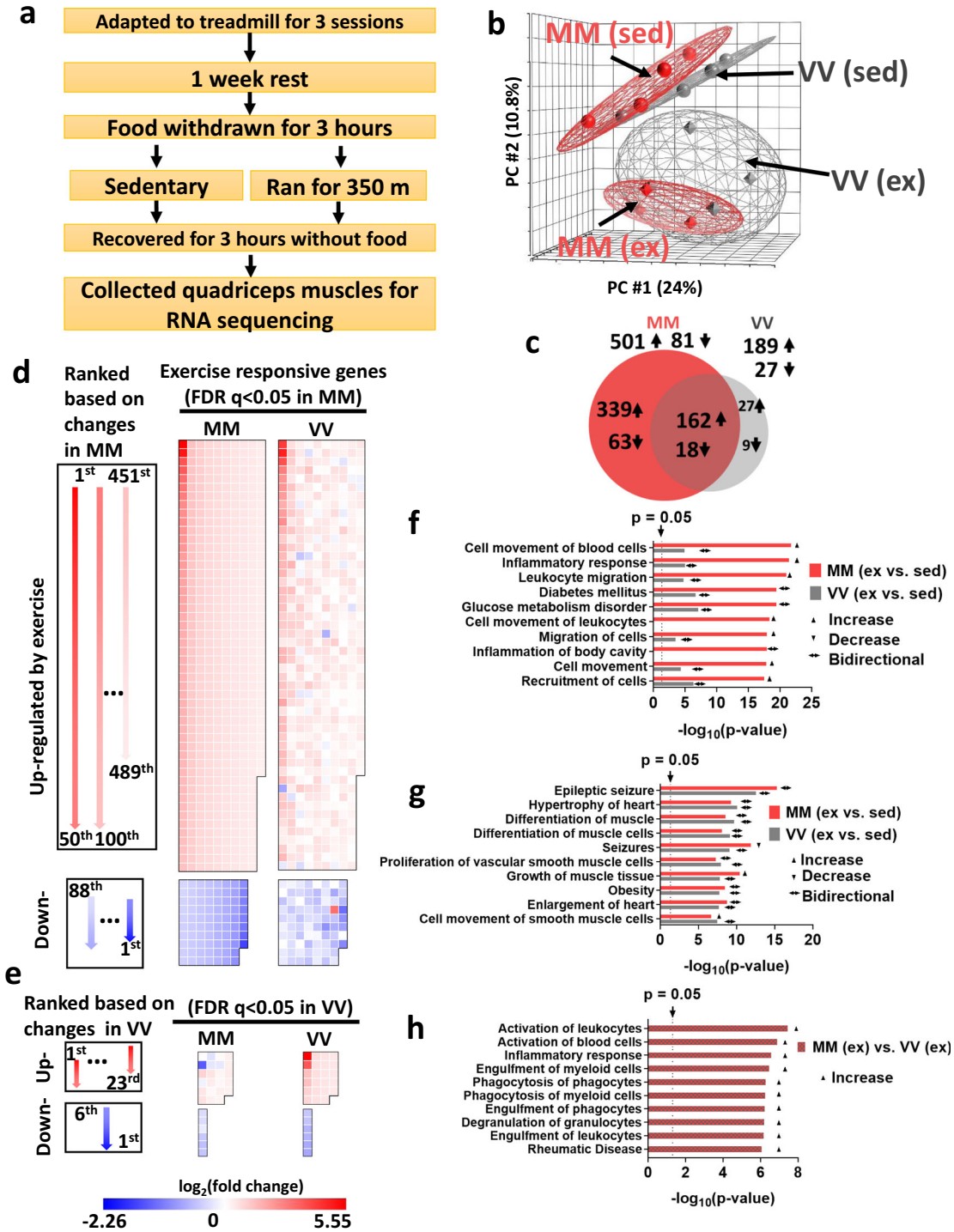

included terms such as "differentiation of muscle cells" and "growth of muscle tissue", which were expected for the adaptive response to exercise[41]. Importantly, the MM (WT) muscles shared a similar degree of pathway enrichment for these biological functions (Fig. 3g). We then directly compared the transcriptomes of exercised MM (WT) and VV muscles to identify a list of genes that showed the most prominent differences (more than ±2σ) between genotypes under this post-exercise condition. When these differentially expressed genes were analyzed by ingenuity pathway analysis, the results (Fig. 3h) further supported the prominent difference in inflammatory responses between exercised MM (WT) and VV

muscles: exercised MM (WT) muscles showed significant enrichments ($P < 0.05$) and activation (z-score ≥2.0) of multiple biological functions related to inflammation (Fig. 3h). Our results suggest that ox-CaMKII plays an important role in coupling ROS to the activation of physiological inflammatory response pathways, a well-established adaptive response to a single bout of unaccustomed exercise[43]. Under disease conditions, CaMKII has been shown to promote inflammation in the heart[44–46] and airway[16], and to function in mast cells[16], macrophages[47–50], and T cells[51]. Our new data established ox-CaMKII as a molecular connection between inflammatory responses and physiological ROS signaling.

**Fig. 3 Ox-CaMKII is important for acute transcriptional responses to exercise. a** Protocol for submaximal exercise and muscle collection. **b** Principal components analysis (PCA) of RNA sequencing results of sedentary (sed) and exercised (ex) muscle samples. The distance between samples in PCA corresponds to similarity (near) or difference (far) in their transcriptional profiles ($n = 4$ mice for each group). **c** Numbers and overlap of significantly changed (false discovery rate-adjusted $q$-value < 0.05) genes in response to exercise in MM and VV muscles. Arrows indicates up- (↑) or downregulation (↓) when comparing exercised muscles to sedentary muscles. **d** Left panel, diagram of the layouts for arranging genes in the middle and right panels; middle panel, genes whose expression was significantly ($q < 0.05$) changed in response to exercise in the MM (WT) muscles are ordered according to the diagram in the left panel, and their $\log_2$ (fold changes) are represented by color; right panel, the $\log_2$ (fold change) of the same genes in response to exercise in the VV muscles are shown. **e** Left panel, diagram of the layouts for arranging genes in the middle and right panels; right panel, genes whose exercise responses to exercise reached significance ($q < 0.05$) only in the VV muscles (excluding seven transcripts not consistently detected across samples due to low expression values), and these genes are ordered according to the diagram in the left panel, and their $\log_2$ (fold changes) are represented by color; middle panel, the $\log_2$ (fold change) of the same genes in response to exercise in the MM muscles are shown. **f** Top ten most significantly (smallest $P$ values) changed functions comparing transcriptomes of exercised MM muscles to their sedentary counterparts identified by Ingenuity Pathway Analysis. Corresponding enrichment $P$ values of the same functions in the exercised VV muscles are plotted for comparison. **g** Top ten most significantly (smallest $P$ values) changed functions comparing transcriptomes of exercised VV muscles to their sedentary counterparts identified by ingenuity pathway analysis. Corresponding enrichment $P$ values of the same functions in the exercised MM muscles are plotted for comparison. **h** Ingenuity pathway analysis directly comparing transcriptomes of exercised MM and VV muscles. In **f**–**h**, $P$ values are from the right-tailed Fisher's exact tests performed by ingenuity pathway analysis, not adjusted for multiple tests. Activation, depression or bidirectional changes of the biological functions are determined by the $z$-score of ingenuity pathway analysis for each pathway ($z \geq 2.0$ for activation, $z \leq -2.0$ for depression, otherwise for bidirectional changes). Source data are provided in the Source Data file.

Similar to exercise, degranulation of mast cells induces gene expression profile changes[52], and mast cell degranulation in response to FcεRI ligation was attenuated by mutating the MM residues of CaMKIIδ to VV[16]. The WT (MMδ) mast cells had larger cytosolic $Ca^{2+}$ transients than VVδ cells[16], mirroring the observation that ox-CaMKIIγ potentiates $Ca^{2+}$ transients in skeletal muscle fibers (Fig. 2h–j). Because transcriptional regulation is an important output of cellular signal transduction, we tested whether ox-CaMKIIδ contributed to gene expression in mast cells after their degranulation. We first compiled a list of 90 candidate genes that might be under the control of ox-CaMKIIδ in mast cells after degranulation (Fig. 4a and Supplementary Data 1). These candidate genes were selected based on two criteria (Fig. 4a): one, they were differentially regulated by exercise between MMγ and VVγ muscles, indicating that they were ox-CaMKIIγ dependent; and two, their orthologs were expressed in human mast cells and were significantly regulated by degranulation[52]. We predicted that if these genes are regulated by ox-CaMKIIδ in mast cells, their expression would be different between degranulated MMδ (WT) and VVδ mast cells. Principal component analysis of RT-qPCR measurements of these genes showed that their expression could not distinguish MMδ (wild-type) from VVδ mast cells prior to degranulation (Fig. 4b). In contrast, their expression clearly distinguished degranulated MMδ (WT) mast cells from VVδ mast cells and showed that MMδ (WT) cells had stronger responses to degranulation (Fig. 4b). These results suggest that like ox-CaMKIIγ, ox-CaMKIIδ also plays important roles in regulating physiological gene expression.

The finding that ox-CaMKIIδ and ox-CaMKIIγ regulate gene expression suggests that this function might be gained by the ancestral vertebrate CaMKII because phylogenetic analysis indicated that CaMKIIδ and CaMKIIγ diverged at the first duplication event of the ancestral CaMKII[53]. To further test the possibility that bridging ROS to gene expression was one of the primordial functions of ancestral vertebrate CaMKII, we turned to the extant phylogenetic bracket established by MM flies and mice. Many mammalian transcription regulators targeted by CaMKII have orthologous counterparts in *Drosophila melanogaster* (http://flybase.org/). We fed MM and VV (WT) flies food with 10 mM paraquat for 24 h at 25 °C, a regimen that induced elevated mortality in MM flies (Fig. 1c), and is known to elicit a stereotyped transcriptional response in *Drosophila melanogaster*[54]. We focused on a subset of the paraquat-induced fly genes[54] whose paralogues in mice were altered by exercise. We selected some of these genes and confirmed by RT-qPCR that all were significantly regulated by paraquat in MM and VV flies (Fig. 4c). Strikingly, none of these genes showed a difference in expression between MM and VV flies consuming control food, while a subset exhibited significant differences between MM and VV flies after paraquat feeding (Fig. 4c). The results showed that introducing the MM module to fly CaMKII bridges ROS to the expression of a specific set of genes, consistent with the concept that acquisition of MM in the ancestral vertebrate coupled ROS to gene expression.

**MM insertion to *Drosophila melanogaster* CaMKII recapitulates antagonistic pleiotropy.** We next considered that if the MM module confers a performance benefit to flies, as it does for mice, then the cellular context necessary to produce such physiological benefits is likely homologous between these model organisms[22], and would have been in place when MM was introduced to the CaMKII regulatory domain of stem vertebrates. We placed the flies into vertical racetracks to measure climbing velocity (Fig. 5a). The MM flies climbed at a significantly higher velocity than VV (WT) flies (Fig. 5b, control condition). The superior climbing performance conferred by the MM module was dependent on the physiological redox state, because ingesting food supplemented with NAC dose-dependently reduced the performance of MM but not VV (WT) flies (Fig. 5b, NAC-treated conditions). The MM flies climbed at similar velocities to VV (WT) flies after treatment by NAC (Fig. 5b). Since climbing performance could be affected by body weight/size, we measured the thoracic length of the flies as a surrogate for body size and found no difference between MM and VV flies (Supplementary Fig. 16a). Another factor that could influence climbing performance is metabolism[55]. Interestingly, MM flies had a significantly smaller triglyceride store (Supplementary Fig. 16b), suggesting that CaMKII activity might promote lipolysis and fatty acid utilization in flies to favor motor performance, similar to its suggested roles in mammalian cells[56,57]. To ascertain that muscles are directly involved in the superior performance of MM flies, we further evaluated the performance of denervated hearts in MM and VV (WT) flies (Fig. 5c). The hearts of MM flies had superior performance, evidenced by significantly higher shortening velocity and relaxation rates (Fig. 5d, e), but the performance advantage of MM hearts diminished in the presence of NAC (Fig. 5d, e).

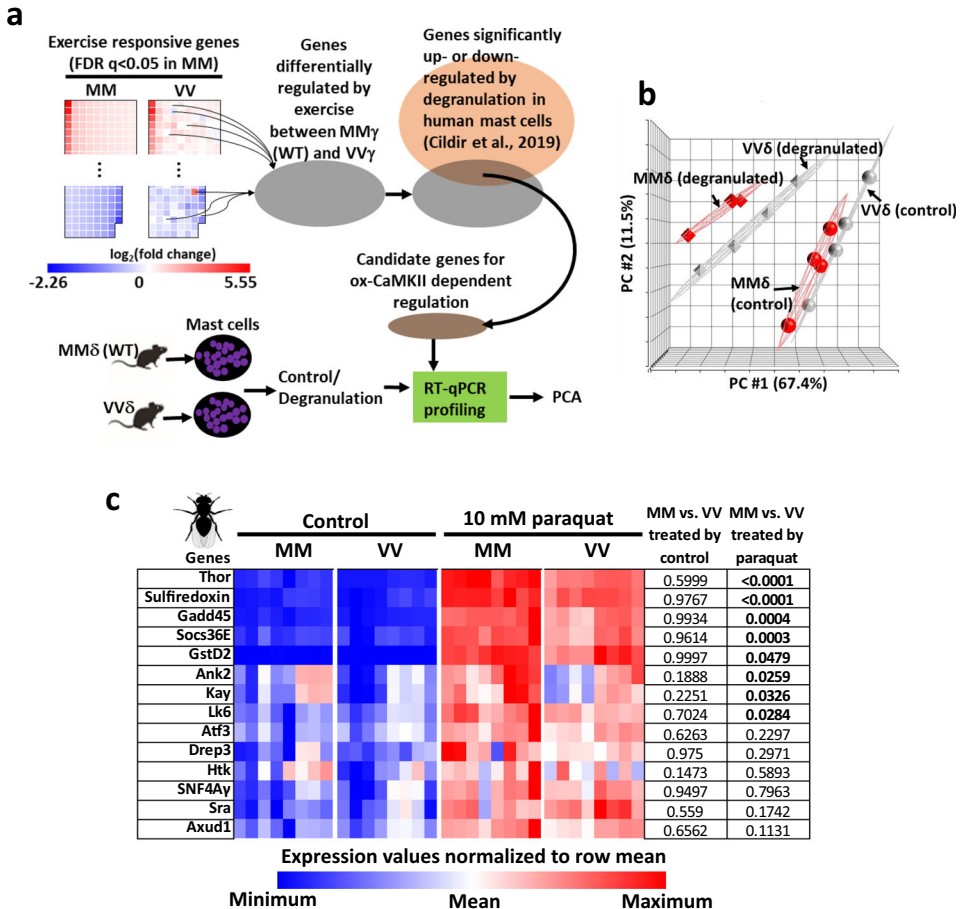

**Fig. 4 The MM modules of CaMKIIγ and CaMKIIδ in mice and of *Drosophila melanogaster* CaMKII couple ROS to gene expression. a** Scheme for identifying candidate ox-CaMKII-regulated genes from exercising skeletal muscle for testing in mast cells. **b** Principal component analysis of expression profiles of genes in wild-type (MMδ) and VVδ mast cells. The mast cells were sensitized with ovalbumin (OVA)-specific IgE for overnight and then treated by vehicle (control) or OVA and harvested 4 h later for RT-qPCR analysis ($n = 4$ per condition). **c** Expression of a subset of paraquat responsive genes was quantified by RT-qPCR after RNA was extracted from flies ingesting control food (Control, 5% sucrose solution) or paraquat (10 mM of paraquat in 5% sucrose solution) for 24 h at 25 °C ($n = 8$ biological replicates per group, each containing 15 males and 15 females). All of these genes were significantly upregulated by paraquat ($P < 0.05$, two-way ANOVA). No genes showed significant differences between MM and VV flies fed control food, whereas a subset of genes had significantly higher expression in MM than in VV flies after exposure to paraquat ($P$ values shown in the table. Sidak's multiple comparisons test, two-tailed). Source data are provided in the Source Data file.

Frailty is a condition associated with pathological aging, linked to maladaptive actions of ROS[7,8], and a plausible consequence of antagonistic pleiotropy[10]. Based on this, we examined the effects of a sublethal dose of paraquat (4 mM for 24 h) on climbing (Fig. 6a). After paraquat treatment, the MM flies exhibited significantly reduced climbing velocity, whereas VV (WT) flies were unaffected (Fig. 6a). We further examined the effects of very low dose paraquat feeding (1 mM for 3–6 days) on spontaneous ambulatory activity. We found that there was no difference in spontaneous ambulation between MM and VV (WT) flies at baseline, whereas exposure to food containing paraquat significantly reduced daily ambulatory activity counts only in MM flies (Fig. 6b). Similarly, the benefits of the MM module on the performance of denervated hearts were completely abrogated when the fly hearts were exposed to 10 mM paraquat for 90 min (Fig. 6c, d); and longer exposure (150 min) to paraquat disrupted the contraction of significantly higher portions of MM than VV hearts (Fig. 6e and Supplementary Movie 1). Taken together our inferential and experimental data align to support a view that ox-CaMKII, by virtue of the MM module, behaves according to the precepts of antagonistic pleiotropy, likely since its emergence in ancestral vertebrates.

## Discussion

Antagonistic pleiotropy has been proposed as a major, and possibly ubiquitous, mechanism for aging-related diseases[6,58]. Well-documented examples of genes that meet the theory's predictions of late-stage detrimental effects coupled with benefits to early life, however, are growing but still surprisingly few[6]. This may be because the fitness benefits of such alleles result in their rapid fixation, thereby eliminating the polymorphism that helps reveal their effects and eases their identification[5]. We overcame this difficulty with an integrated methodology that combined phylogenetic and experimental approaches. Our studies provide new in vivo and in vitro evidence that ox-CaMKII is an important antagonistic pleiotropic genetic trait that confers important fitness benefits, while also promoting disadvantageous sensitivity to ROS that contributes to aging-related diseases, and possibly to frailty. We found that ox-CaMKII directly orchestrates connections between ROS, intracellular $Ca^{2+}$, and gene transcription that lead to physiological advantages in mice. Parallel experiments in *Drosophila*, combined with pattern-based phylogenomic data, support the inference that similar physiological benefits would have accompanied ox-CaMKII when it first evolved on the vertebrate stem lineage. The fixation of the MM/CM oxidative

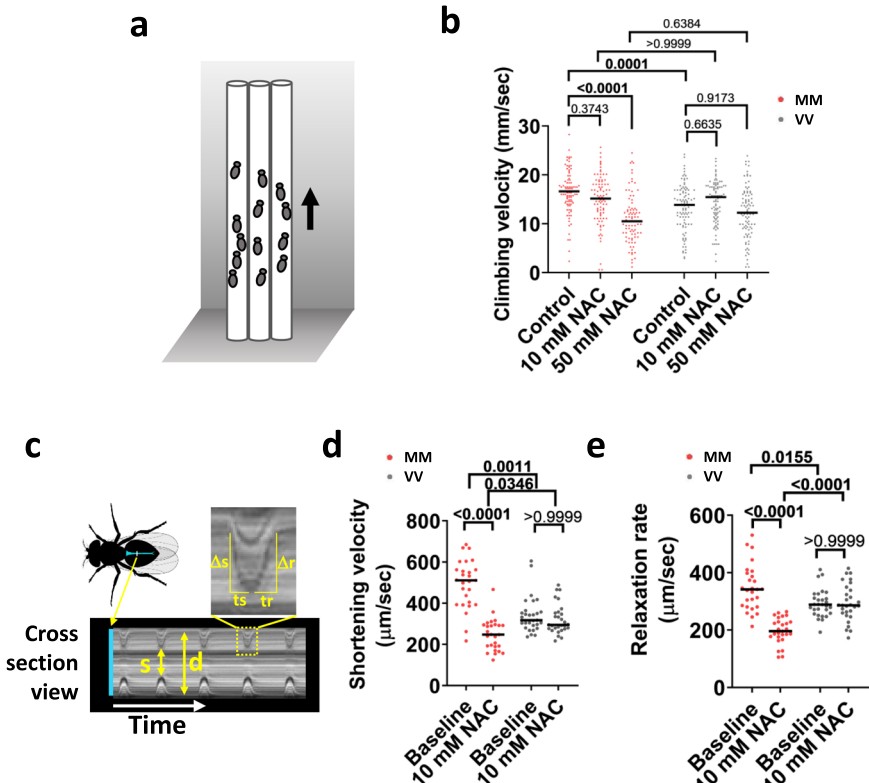

**Fig. 5 MM module couples ROS to improved performance in *Drosophila melanogaster*. a** Diagram for the climbing test. **b** Vertical climbing velocity of flies treated by control food (5% sucrose) or food containing 10 mM or 50 mM NAC for 24 h; $n = 93$ control-treated MM, $n = 90$ 10 mM NAC-treated MM, $n = 86$ 50 mM NAC-treated MM, $n = 94$ control-treated VV, $n = 91$ 10 mM NAC-treated VV, $n = 94$ 50 mM NAC-treated VV. Horizontal lines indicate medians. *P* values shown above the brackets, Dunn's nonparametric multiple comparisons tests, two-tailed. **c** Diagram of a fly heart (in blue color) and an example kymograph recording of heart dynamics. Arrows indicate systole (s) and diastole (d) of the heart. The boxed region of the kymograph was magnified to show that the shortening velocity was derived from Δs/ts, and the relaxation rate was derived from Δr/tr. **d** and **e** Cardiac performance indices of MM and VV hearts before and after 60-min treatment by 10 mM NAC. The shortening velocity **d** and relaxation rate **e** were assessed; $n = 27$ MM hearts per condition and $n = 29$ VV hearts per condition; horizontal lines indicate medians; *P* values shown about the brackets in **d** and **e**, two-tailed tests; Dunn's nonparametric multiple comparisons tests for **d**, and Welch's ANOVA test followed by Dunnett's T3 multiple comparisons test for **e**; choice of tests was determined by data normality. Source data are provided in the Source Data file.

pathway among extant vertebrates represents strong evidence that the underlying sequences are not evolving randomly, as predicted under a mutation accumulation model, and are likely maintained by positive selection. A well-supported secondary conclusion is that ox-CaMKII was a key innovation in facilitating the heightened physiological output required of these derived anatomical systems and thus played a critical role in the initial establishment and continued evolutionary success of vertebrates.

Our observations and experiments indicate the MM/CM motif as a decisive, adaptation that established CaMKII as a conduit for oxidant stress to engage cytoplasmic $Ca^{2+}$, gene transcription, cell survival[11,13,16,17], and membrane excitation[12]. Together, these processes appear poised to collaborate to enhance physiological performance, but, unfortunately, contributed substantial vulnerability to promote ROS-triggered toxicity and disease (Fig. 7). Thus, ox-CaMKII provides a unique example for understanding the hormesis effects of ROS and suggests that beneficial and detrimental aspects of oxidative stress can be transacted by the same molecular mechanism (Fig. 7). While excessive oxidant stress is implicated in essentially all major diseases, to date there is scant evidence that antioxidant therapies are effective in preventing or treating disease[59]. Our findings suggest successful antioxidant therapies will require improved understanding, and more precise targeting of ROS engaged molecules and pathways. It is worth noting that this and a growing number of studies

consistently linked CaMKII activity to inflammation[16,44–50]. Inflammation is a central feature of aging[60], our finding that CaMKII oxidation can drive inflammatory gene expression suggests that ox-CaMKII might be a direct link between aging-associated oxidative stress and aging-associated inflammation. Due to the presence of four *CaMK2* paralogs in mammals, all of which are ROS-sensitive, the relative and combined roles of CaMKII isoforms in mammalian aging require further study.

The conservation of the MM/CM module in essentially all isoforms of vertebrate CaMKII further suggests that ox-CaMKII plays diverse physiological roles, beyond those uncovered by this study in the skeletal muscles and mast cells. The formative evolutionary role of the MM/CM module is paired with considerable irony, given the well-recognized contributions of ox-CaMKII to major chronic and life-threatening human diseases. The striking observation that the MM/CM module enacts the performance/disease trade-off in flies, whose most recent common ancestor with vertebrates lived well over half-a-billion years ago is particularly worth noting. It suggests that the MM/CM module is a concise but highly impactful ROS sensor, and once it was obtained by CaMKII in the stem vertebrate, the MM/CM module was sufficient to couple ROS to a wide range of CaMKII targets important for enhanced performance, gene expression, disease, and death. The fact that ox-CaMKII can confer a performance advantage to flies, but ox-CaMKII is absent in invertebrate

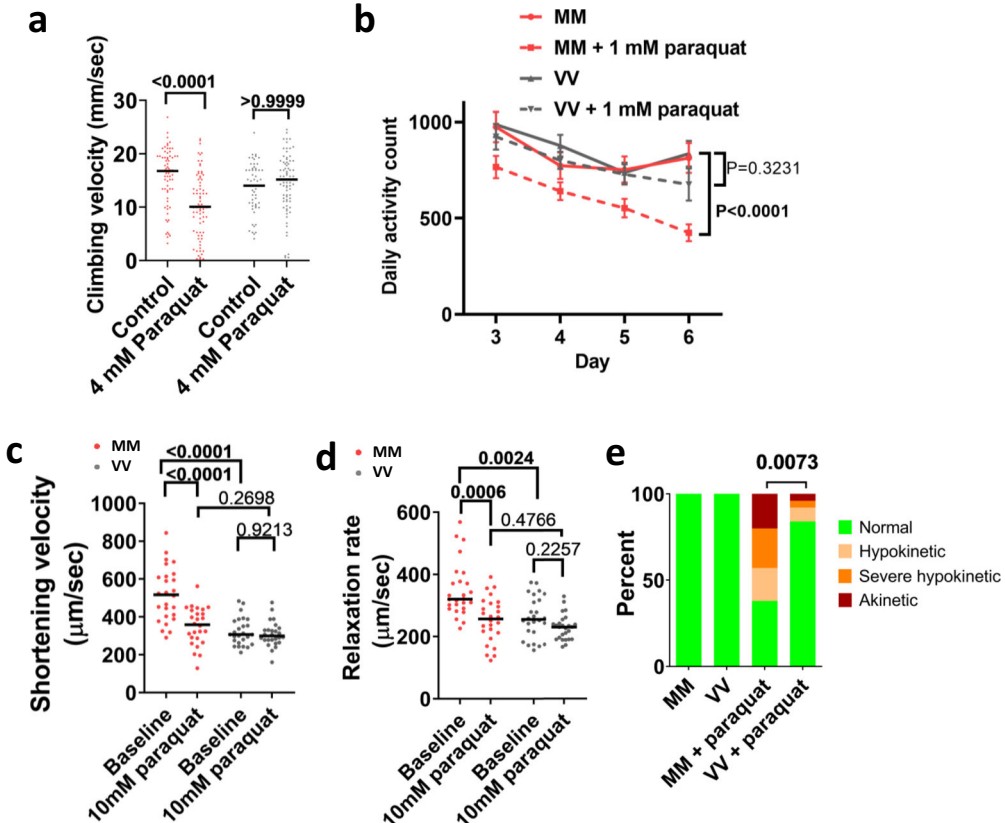

**Fig. 6 The MM module sensitizes flies to pathological oxidative stress. a** Vertical climbing velocity of flies treated by control food or food containing 4 mM paraquat for 24 h at 25 °C; $n = 68$ control-treated MM, $n = 74$ paraquat-treated MM, $n = 57$ control-treated VV, $n = 75$ paraquat-treated VV flies. Horizontal lines indicate medians. P values from two-tailed Dunn's multiple comparisons tests shown in the graph. **b** Daily activity counts of MM and VV flies consuming a control or paraquat (1 mM) diet. Diets started at day 1 and behavior monitoring occurred between days 3 and 6 (points and error bars are mean ± SEM. P values calculated using two-tailed Tukey's multiple comparisons test for the effect of paraquat within the genotypes, $n = 29$ control MM, $n = 21$ paraquat-treated MM, $n = 31$ control VV, and $n = 21$ paraquat-treated VV flies). **c, d** Cardiac performance of hearts bathed first in control artificial hemolymph and then in hemolymph containing 10 mM of paraquat for 90 min; $n = 26$ hearts per genotype; horizontal lines indicate medians; P value from two-tailed Dunnett's T3 multiple comparisons tests for **c** and two-tailed Dunn's multiple comparisons tests for **d**; choice of tests was determined by data normality. **e** All MM and VV hearts showed normal contraction before paraquat treatment, however, after exposure to 10 mM paraquat for 150 min, significantly more MM hearts became hypokinetic, severely hypokinetic, or akinetic (examples of categorical cardiac performance are in Supplementary Movie 1). $n = 26$ hearts per genotype, P value from Chi-square test. Source data are provided in the Source Data file.

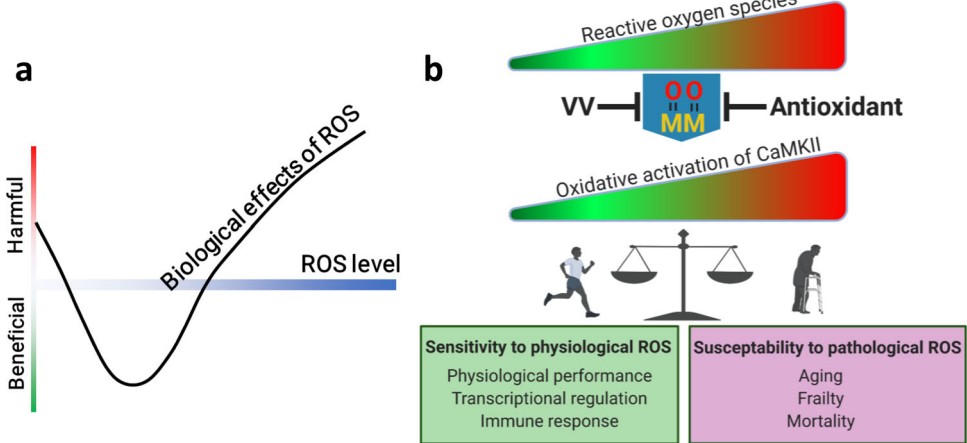

**Fig. 7 The double-edged roles of CaMKII oxidation. a** ROS induces a hormesis dose response, where too little or too much ROS are harmful, and optimal biological effects are achieved with an intermediate level of ROS. **b** ROS dictates the activity of CaMKII by oxidizing the MM/CM module. A physiological level of ROS allows CaMKII activity required for optimizing physiological functions, which is the likely reason for the evolution of the MM/CM module. Excessive oxidative stress induced by disease and aging hyperactivates CaMKII, which in turn aggravates frailty, diseases, and aging. Blocking oxidative activation of CaMKII by mutating MM residues to VV or by antioxidants protects against pathological processes accompanied by excessive oxidative stress, but also negates the physiological benefits of CaMKII oxidation. Diagrams were created with BioRender.com.

lineages suggests that the detriments of ox-CaMKII outweigh its benefits in invertebrates. Possibly, invertebrates may not be equipped to regulate CaMKII oxidation effectively. Furthermore, invertebrates may lack the full suite of physiological traits, such as adaptive immunity, to fully benefit from ox-CaMKII. The totality of our study strongly supports the conclusion that the MM/CM module is an evolutionary trade-off in vertebrates that uses ROS to enhance physiological performance, while simultaneously bestowing sensitivity to ROS for promoting multiple, common chronic diseases, many of which transpire late in life, beyond the reach of natural selection[3,5].

## Methods

**Animal use**. All animal handling procedures were in accordance with National Institutes of Health guidelines and were approved by the Institutional Animal Care and Use Committees of Johns Hopkins University School of Medicine. All mice were housed in the Johns Hopkins animal facility under 12-h light, 12-h dark cycles and fed 2018 Teklad global 18% protein rodent diets ad libitum. The room temperature was kept at $22 +/- 1\,°C$, and humidity at $40 +/- 10\%$. We used only male mice in this study because most of the studies supporting the role of the MM module in contributing to disease severity were performed in male mice, and because there is a large difference in physiological performance between males and females. Based on these considerations, we decided that it would be appropriate to focus on male mice to evaluate the role of MM-CaMKII in physiological performance. We acknowledge the possibility that ox-CaMKII may have different effects in females, especially considering that females and males have different susceptibility to aging-related diseases.

All flies were cultured in polypropylene, wide fly *Drosophila* vials (Genesee Scientific, #32–121) containing Nutr-Fly BF food (Genesee Scientific, #66–112) at 25 °C. To produce flies for experiments, we set up crosses with eight females and four males in each vial for 2 days.

We used both males and females for most fly experiments. However, the climbing assay accompanying heart tube performance assays, and the spontaneous activity assay were performed using only females. We chose females instead of males for these assays because males sleep significantly more than females during the day[61], which may confound assays related to motor functions.

**Comparative modeling of *Drosophila melanogaster* CaMKII structure**. Details of modeling are presented in Supplementary Fig. 3. In addition to evidence from modeling suggesting that MM- fly CaMKII is a hyperactive enzyme in the presence of ROS, we recently reported a different line of flies harboring a hypomorphic CaMKII mutation[62]. Flies with the hypomorphic CaMKII show reduced activity and exhibit a striking reduction in heart tube contractility[62], the opposite of the phenotype we observed in MM flies without NAC treatment. Taken together, these findings provide strong support that the MM module augments CaMKII activity in the presence of ROS.

**Generation of *CaMKII^MM^* point mutation in *Drosophila melanogaster* by CRISPR-mediated gene editing**. The genomic sequence of the *Drosophila melanogaster CaMKII* gene was used in the CRISPR guide design tool (http://crispr.mit.edu/) to create the CRISPR guides. Guide #1 (GTTACAGCAACGCGAACGTG) was chosen due to its close proximity to the codons encoding V281 and V282 and its lack of high probability off-targets. Guide #1 was ordered as complementary oligomeric DNA (Integrated DNA Technologies) and cloned into the pU6-BbsI-ChiRNA plasmid[46] (the pU6-BbsI-chiRNA was a gift from Melissa Harrison & Kate O'Connor-Giles & Jill Wildonger (Addgene plasmid # 45946; http://n2t.net/addgene:45946; RRID:Addgene_45946)). A single strand 176 nt ultramer DNA oligo (ssODN-1R) was designed as the template for HDR-mediated point mutations and was ordered from Integrated DNA Technologies (Supplementary Fig. 2). The sequence of ssODN-1R is "AACATTGTCGTAAGTATGGCTCCCTTTAGC TTGCGCGCGCATTAAATTTCTTGAGACAGTCTACGGTTTCTTGGCGATG CATCATGGAAGCGACTCGTTCGCGTTGCTGTAACAATGTTTTTTCATTAT CTTTATGTAAACCTAAGAGAAAAATTAGTCTGCACTTACACAAATC". Injection of the ssODN-1R and the guide RNA encoding plasmids into fly embryos was carried out by Rainbow Transgenic Flies, Inc (3251 Corte Malpaso Unit 506 Camarillo CA). Genotyping was carried out by PCR amplification from genomic DNA, extracted from the wings of the flies, with primer-F (GTCGGTTATC-CACCCTTTTG), and primer-R (GACGCCAAGTATATTGATGTGG) followed by Sanger sequencing and NsiI/NsiI-HF digestion. The flies with the correct *CaMKII^MM^* allele were backcrossed with iso31 flies[63] for five generations to minimize the possibility of carrying off-targets from the CRISPR-mediated gene editing.

**Phylogenetic survey of CaMKII**. Most CaMKII orthologues listed in Supplementary Fig. 1 and Supplementary Table 1 were identified in the Interpro database (http://www.ebi.ac.uk/interpro/entry/IPR013543/taxonomy), based on the criteria

that the sequences have the conserved CaMKII association domain, and a kinase domain. Additional sequences were uncovered by BLAST in the NCBL nucleotide database and translated into proteins (https://blast.ncbi.nlm.nih.gov/Blast.cgi). All sequence files with accession numbers are available in Supplementary Data 2. The CaMKII sequences were aligned using the Molecular Evolutionary Genetics Analysis software MEGA-X[64] (https://www.megasoftware.net/). The evolution of an oxidation-sensitive amino acid pair at loci 281/282 of the CaMKII regulatory domain is an unambiguous synapomorphy of crown-clade vertebrates among Deuterostomata. The initial identity of this pair was CM, with the 281 cysteine likely being subsequently replaced by a methionine in one of the two paralogs that resulted from a full round of genome duplication that occurred prior to the origin of the vertebrate crown clade. Our phylogenetic survey did recover a small number of non-deuterostome taxa that also exhibit oxidizable residues at those same regulatory loci (Supplementary Table 1). The large phylogenetic separation between these taxa, both individually and collectively, from Deuterostomata leaves it clear that they evolved independently of the vertebrate condition; they are also MM rather than the CM of the earliest vertebrates. As a greater taxonomic diversity of metazoan genomes become available, a meaningful probabilistic analysis of ox-CaMKII evolution outside of Deuterostomata will be possible. But that analysis is highly unlikely to question the evolutionarily unique nature of vertebrate ox-CaMKII.

**Generation of *CaMKII^VV^* knock-in mutation in mice**. *CaMKII^VV^* knock-in mice were generated by GenOway (https://www.genoway.com/) with mouse embryonic stem cells of the C57BL/6 background as specified in Supplementary Fig. 7. The mice used in experiments had been further backcrossed to C57BL/6 J mice (The Jackson Laboratory, 000664) and all experiments were carried out with littermates.

**Ncf1^−/−^ mice**. Ncf1^−/−^ mice were purchased from the Jackson Laboratory (Cat #004742) and maintained on a C57BL/6J background.

**Western blotting for mitochondrial complexes**. Protein extracts were prepared from frozen tissue with T-PER Tissue Protein Extraction Reagent (Thermo Scientific, #78510) in the presence of protease (Sigma-Aldrich, P8340) and phosphatase (Sigma-Aldrich, P0044) inhibitors. The mice were 14-week-old at the time of sacrifice. The primary antibodies were the Total OXPHOS Rodent WB Antibody Cocktail from Abcam (ab110413, 1:1000), and GAPDH (D16H11) XP® Rabbit mAb (Cell Signaling, #5174, 1:2000). Data were collected by LI-COR Odyssey Fc (Lincoln, Nebraska 68504, USA).

**Skeletal muscle fiber typing**. Skeletal muscle fiber composition was determined by immunostaining following a standard method[65] from a cohort of 14-week-old mice The primary antibodies BA-F8-c (myosin heavy chain, slow, 1:50), SC-71-c (Myosin Heavy Chain Type IIA, 1:600), BF-F3-c (Myosin Heavy Chain Type IIB, 1:100), 6H1-s (myosin heavy chain, fast, IIX, 1:50) were obtained from Developmental Studies Hybridoma Bank (University of Iowa).

**Paraquat, and NAC treatment and behavior study of flies**. To determine the effects of paraquat on mortality, newly eclosed flies were sorted under $CO_2$ anesthetization and placed into individual vials, and each vial received ten males and ten females. When the flies reached 5–7 days old, they were transferred into vials containing a filter paper pad (cut from Bio-Rad #1704085) soaked with 600 μL of 5% sucrose solution (control) or 5% sucrose solutions containing 10 mM, 25 mM or 50 mM paraquat (Sigma-Aldrich, # 856177). Mortality of flies was recorded at 24 and 48 h after initiation of the treatment.

To test the negative geotaxis (climbing), females were collected and aged, as above, and kept in vials in groups of ten flies; we tested six to ten groups per condition in the geotaxis assay, resulting in sample sizes between 60 and 100. For paraquat or NAC treatment prior to the negative geotaxis test, the flies were treated with 5% sucrose, 5% sucrose + 4 mM paraquat, 5% sucrose + 10 mM NAC, or 5% sucrose + 50 mM NAC for 24 h at 25 °C. They were then transferred into vertical test tracks made from 25-mL serological pipette tubes using a funnel without anesthetization. During the climbing test, the flies were dislodged to the bottom of the tubes by rapidly tapping the vials on the desktop ten times and climbing was video recorded for subsequent analysis. Each group of flies was tested for ten consecutive trials at 30 s intervals. The vertical distances the flies climbed in 6 s since the last tap (time 0) were used to calculate the vertical velocity of climbing. Flies that initiated flight or paused during the 6-s time window were excluded from the analysis. Because of this exclusion criterion, the final sample sizes were different from multiples of ten, and the specific *n* is shown in the figure legends. We found that the flies performed reproducibly from the second to tenth trials and presented data from trial 2 in Figs. 5b and 6a.

To determine the effects of very low dose paraquat (1 mM) on daily ambulatory activity, individual 1-week-old female flies were anesthetized by $CO_2$ and loaded into tubes containing control or paraquat-containing food and monitored by *Drosophila* Activity Monitoring System (Trikinetics). Fly behavior was recorded from day 3 to 6.

**Lifespan of flies**. Newly eclosed flies were collected under $CO_2$ anesthetization and housed in 6 oz flasks containing 25 ml of standard food at a density of about 330 flies per flask. For 4 days, the flies were allowed to mature and copulate. On day 4, they were anesthetized by $CO_2$, sorted into males and females, and placed into individual vials as groups of 30 flies. The vials were placed in environmental chambers at 25 °C or 29 °C with 12-h light/12-h dark cycles. The flies were counted and flipped into fresh vials every 2 days until all flies in the vials died. The mortality data were analyzed in GraphPad Prism; a few flies that escaped during handling were excluded from the analysis.

**Measurement of the thoracic length of flies**. Newly eclosed female flies were housed at a density of 30 flies per vial containing standard food until 5 days post eclosure. The flies were anesthetized by FlyNap (Catalog #173010, Carolina Biological Supply Company, Burlington, NC) and imaged under a Leica M165 FC stereomicroscope. The images were analyzed in Fiji ImageJ[66] to derive the thoracic length.

**Triglyceride content of flies**. Newly eclosed female flies were housed at a density of 30 flies per vial containing standard food until 5 days post eclosure. The flies were harvested and randomly assigned into groups of eight flies for sample processing and measurement of triglyceride content with a commercial kit (Triglyceride Assay Kit—Quantification, catalog # ab65336, Abcam). Each group was homogenized in 1 ml of lysis buffer containing 5% NP40 and processed according to the kit protocol. 25 μl of lysate from each sample was measured for triglyceride content by colorimetric approach in 96-well plates. Readings from wells that omitted the lipase were subtracted as background from corresponding samples.

**Aconitase activity of flies**. Newly eclosed flies were sorted into groups of 30 flies (15 males and 15 females per vial) and cultured until 4 days post post eclosure. They were then treated with either 5% sucrose solution (control) or 5% sucrose solution plus 4 mM paraquat for 24 h. The flies were then snap-frozen in liquid nitrogen and saved at −80 °C. We measured aconitase activity in the flies with an Aconitase Activity Assay kit (Catalog # MAK051, Sigma-Aldrich) following the kit protocol. Briefly, each group of 30 flies was homogenized in 400 μl of Assay Buffer, sonicated for 20 s, and then centrifuged at $20,000 \times g$ for 15 min at 4 °C. We measured the aconitase activity in 8 μl of lysate from each sample.

**Fecundity tests**. To assess the lifetime fecundity of females, each newly eclosed female and two 4-day-old males were placed into a vial for 6 days at 25 °C. The female was then transferred to a new vial together with two new 4-day-old males. Although female flies store sperms and do not require additional copulation to remain productive, we kept the presence of young males throughout the experiment to keep the experimental condition constant. The new generation of flies was tallied from the vial on the 11th day after the egg-producing female was transferred away into a new vial. This procedure was repeated every six days until the female died or stopped producing eggs.

**_Drosophila_ cardiac physiological analysis**. Dorsal cardiac tubes of 10-day-old female _CaMKII^WT_ (denoted VV (WT)) and _CaMKII^MM_ (denoted MM) flies were dissected in oxygenated artificial hemolymph[67]. Myogenic contractions of cardiac tissue were recorded using the Hamamatsu Orca Flash 2.8 CMOS camera on a Leica DM5000B TL microscope with a ×10 immersion lens at ~120 frames per second at baseline and after 90 and 150 min following the addition of 10 mM paraquat, or 60 min after the addition of 10 mM NAC. Cardiac physiological indices were determined using the semi-automated optical heartbeat analysis program[68,69]. Significant differences between genotypes before and after paraquat treatment for 90 min were determined by two-tailed Mann–Whitney tests. After 150 min in paraquat, many of the MM hearts no longer contracted and the cardiac indices could not be meaningfully derived. We, therefore, categorized the contractions as normal, hypokinetic (part of the heart contracting), severe hypokinetic (only twitching could be observed in part of the heart), and akinetic (no movement). Representative videos for each category are shown in Supplemental Video 1. The categorical data were assessed using a Chi-square test.

**Construction and validation of a CaMKII activity sensor CaMKII-KTR**. We were unable to validate existing stocks of ox-CaMKII antiserum[11], and developed the novel CaMKII activity reporter, CaMKII-KTR in order to measure dynamic CaMKII activity in living, electronically stimulated skeletal myocytes. The CaMKII-KTR was constructed based on the principles previously published[31] and described in Supplementary Fig. 9. Specifically, the sensor consists (from N-terminus to C-terminus) of a CaMKII-binding region derived from HDAC4, a linker, a nuclear localization signal (NLS), a nuclear exporting signal (NES) and a fluorescent protein. Optimized CaMKII phosphorylation sites were built into the NLS and NES while keeping the NLS and NES functional. The protein sequence of the CaMKII-KTR, excluding the enhanced green fluorescent protein, is EQELLFRQQALLLEQ QRIHQLRNYQASMEAAGIPVSFGSHRPLKRTASVNEDEAPSKKPLARTASVSS RLERLTLQSS. A cDNA encoding this sequence was ordered as a codon-optimized gene block (gBlock_HDAC4-NLS-NES) from Integrated DNA Technologies

(gccaccatgGAACAGGAACTGCTCTTCCGGCAACAGGCACTTCTGTTGGAGC AGCAACGAATCCATCAACTTAGAAACTACCAAGCATCAATGGAAGCAGC CGGGGATTCCTGTCTCCTTCGGATCTCACAGACCTCTCAAAAGGACAGCT AGTGTAAACGAGGACGAAGCACCTTCAAAGAAACCCTTGGCTAGGACC GCTAGTGTCAGTAGTCGACTGGAGCGGTTGACACTTCAAAGTTCC). The gBlock_HDAC4-NLS-NES was cloned into Cerulean-N1 vector[70] (Cerulean-N1 was a gift from Michael Davidson & Dave Piston, Addgene plasmid # 54742; http://n2t.net/addgene:54742; RRID:Addgene_54742) by In-fusion cloning technology (In-Fusion® HD Cloning Plus CE, Takara, CA). The Cerulean encoding region was then replaced by a stretch of cDNA encoding eGFP, derived from pEGFP-C1 (Takara, CA).

To validate the response of the CaMKII-KTR to the intracellular activity of CaMKII, we transfected (FuGENE® HD Transfection Reagent, Promega, WI) the CaMKII-KTR or co-transfected it with CaMKII, CaMKII^K43M, and CaMKIIN constructs into RPE-1 cells and stimulated the cells with 50 μM histamine. Before cells were imaged, we replaced the culture medium with Live Cell Imaging Solution supplemented with 4.5 g/L glucose (ThermoFisher Scientific, A14291DJ), and stained their nuclei with Hoechst 33342 (ThermoFisher Scientific, #62249) for 20 min to facilitate identification of the nuclei. Fluorescent images were collected using an Olympus IX83 epifluorescence microscope equipped with an ORCA Flash 4.0 sCMOS camera and UPLSAPO20X NA0.75 objective lens using CellSens Imaging Software, V2.3 64 bit, Olympus Scientific Solutions Americas Corp. Waltham, MA, USA. Cells were maintained at 37 °C in an OkoLabs stage top incubator. Image analyses were carried out in CellProfiler[71], which identified the nuclei and five-pixel-wide cytosolic rings surrounding the nuclei. The cytosolic to nuclear KTR signal ratios were calculated using the median intensities measured from the nuclei and cytosolic rings of individual cells. We have recently published further validation of CaMKII-KTR in neonatal cardiomyocytes[72].

**Mouse treadmill exercise**. Exercise capacity tests were carried out with the Exer 3/6 Rodent treadmill (Columbus Instrument, Columbus, OH). Prior to exercise capacity testing, the mice (12–15-week-old) were acclimated to the treadmill for three 10-min sessions on 3 consecutive days. The treadmill was set to 10° inclination and the speed was set to 0, 5, and 10 m/min for the first, second and third acclimation sessions, respectively. The electric shock grid at the rear end of the treadmill was turned on and set at stimulation intensity of 9 and frequency of 3 Hz. During exercise capacity testing, each mouse was placed into a lane of the treadmill. The genotype of the animals was blinded to the operator. The exercise protocol consisted of the following steps: (1) 10 m/min for 2 min for warm-up, (2) continuous acceleration from 15 m/min at a rate of 0.6 m/min² until the mouse was exhausted. Exhaustion was determined when the mouse stayed on the shock grid continuously for 5 s and was determined by the same observer for all experiments. Glucose and lactate were measured from a drop of blood from the tail tip before and immediately after exercise, using a OneTouch Ultra 2 glucometer (Lifescan, Inc) and a Lactate Plus lactate meter (Nova Biomedical), respectively.

**Mouse voluntary wheel running and accompanying metabolic study**. Voluntary wheel running data were collected from male mice (11 weeks of age) tested for 6 days in an open-circuit indirect calorimeter outfitted with running wheels (Comprehensive Lab Animal Monitoring System, CLAMS, Columbus Instruments) at the Center for Metabolism and Obesity Research service core. Data were collected continuously (Oxymax software, v.5.9, Columbus Instruments). Days 1–5 of acclimation to wheel running were monitored for expected daily increases in the number of wheel rotations; the analysis of day 6 is presented. The instrument also provided data for voluntary physical activity in the main cage as indexed by counts of infrared beam breaks, intakes of powdered diet (2018, Envigo), as well as rates of $O_2$ consumption (VO₂, ml/kg/hr) and $CO_2$ production (VCO₂, ml/kg/hr). Oxymax software calculated the respiratory exchange ratio (RER = VCO₂/VO₂) to assess the oxidized fuel mixture being oxidized, and the rates of energy expenditure (EE, kcal/kg/h; EE = VO₂ × (3.815 + (1.232 × RER))). The standard outputs of indirect calorimetry data as per-kg/hr were also renormalized and analyzed as per-kg-lean mass/h.

**Mouse body composition**. Body composition data for lean mass and fat mass were obtained using an EchoMRI-100 at the Johns Hopkins University Phenotyping service core from the same mice undergone the metabolic tests.

**Intraperitoneal glucose tolerance test and insulin sensitivity test**. Male mice between 12 and 13 weeks of age were used for glucose tolerance and insulin sensitivity tests. Intraperitoneal glucose tolerance test was performed following a published protocol[73]. The mice started fasting at 5:30 PM for 16 h and then were injected with 20% (W/V in normal saline) glucose solution intraperitoneally at 2 g/kg. For the insulin sensitivity test, food was withdrawn for 6.5 h and 1 U/kg recombinant human insulin (ThermoFisher Scientific, Catalog # 12585014) was injected intraperitoneally. For both tests, blood glucose was measured from tail blood.

**Assessments muscle function in vivo**. Grip strength measurements were carried out as described previously using a grip strength meter (Columbus Instruments, Columbus, OH, USA) on a cohort of 14-week-old mice[74]. Briefly, a mouse was

suspended by the tail and allowed to grip the horizontal bar of the strength meter. Once a grip is established, the mouse was quickly pulled by the tail until it lost the grip, and the force exerted was recorded. Each mouse performed grip strength test until six successful attempts were accumulated, and the maximal force among the six attempts was taken as the grip strength.

In vivo quadriceps torque measurement was described previously[37]. Briefly, the 10–11-week-old mice were anesthetized under 4% isoflurane and then maintained at 1%. Then their pelvis, torso and femur were stabilized on the apparatus. Afterward, the distal leg was taped to a lever arm, which was connected with a torque cell and positioned at 45° of knee flexion for optimal muscle length. The femoral nerve was stimulated subcutaneously to induce maximal quadriceps muscle contractions and the torque produced was recorded by a connected computer for subsequent analysis. The voltage of the stimulation was optimized prior to the studies to produce the maximal torque.

**In cellulo study of skeletal muscle fibers**. Electroporation of DNA into flexor digitorum brevis (FDB) skeletal muscles of 10–13-week-old mice, muscle fiber culture, imaging of action potential-induced $Ca^{2+}$ transients and cytosol/nucleus distribution of CaMKII-KTR followed our previous reports[32,39]. Fibers were loaded with rhod2 AM (ThermoFisher, catalog # R1244) at 2 µM for 60 min at RT. Fibers were stimulated for 1 h using the protocol shown in Fig. 2g. High-speed rhod2 fluorescence confocal microscopy measurements were carried out before and after fatigue stimulation on a Zeiss LSM 5 Live system and viewed with a 63 × /1.2 NA water immersion objective. Excitation was provided by a 532-nm laser with emission detected using a long-pass 550-nm filter. Field stimulation (square pulse, 20 V × 1 ms) was produced by a custom pulse generator through a pair of platinum electrodes. Images were acquired in line scan mode at 100 µs/line for 1 s using Zeiss LSM 5 live software. The average intensity of fluorescence within selected regions of interest (ROI) was measured with Image Examiner (Carl Zeiss). Images were background corrected by subtracting an average value recorded outside the fiber. The average resting fluorescence (F0) value in each ROI before electrical stimulation was used to scale rhod2 signals in the same ROI as ΔF/F0. For N-acetyl-L-cysteine (NAC) treatment, the fibers were incubated for 20 min with 2 mM NAC (Sigma-Aldrich, St. Louis, MO; catalog # A-7250). Images using CaMKII-KTR were collected using an Olympus FluoView500 confocal system, Olympus IX71 microscope, and UPLAN 20X NA0.75 objective lens. The cytosolic to nuclear CaMKII-KTR signal ratios were calculated using the average intensities measured from two to three nuclei and cytosolic regions of interest of individual cells using Fiji ImageJ. Fibers for $Ca^{2+}$ imaging and KTR studies were examined for structural integrity and twitch responses to field stimulation before being tested.

For the Seahorse study, FDB skeletal muscle fibers from 12- to 13-week-old mice were isolated 1 day before the experiments and plated to a laminin-pretreated Seahorse XF96 Cell Culture Microplates overnight. Cell metabolism and bioenergetic analyses of muscle fibers were performed using an Agilent XF96 Extracellular Flux Analyzer. XF Cell Mitochondrial Stress Test kit and Glycolysis Stress Test kit were used to measure mitochondria respiration capacities and cellular glycolysis capacities following the manufacturer's protocol. In the mitochondrial stress assay, muscle fibers were incubated in the muscle fiber assay medium (120 mM NaCl, 3.5 mM KCl, 1.3 mM $CaCl_2$, 0.4 mM $KH_2PO_4$, 1 mM $MgCl_2$, 5 mM HEPES, and 10 mM glucose, pH 7.4) followed by port injections of the final concentration of 1 µM oligomycin, 0.5 µM FCCP, 10 mM pyruvate and 0.5 µM antimycin A/rotenone. In the glycolysis stress assay, glucose was not included in the initial muscle fiber assay medium, and then 2 mM glutamine, 10 mM glucose, 1 µM oligomycin, and 50 mM 2-DG were injected sequentially. The oxygen consumption rate (OCR) and extracellular acidification rate (ECAR) were analyzed using Seahorse Wave software. OCR and ECAR were normalized by the number of skeletal muscle fibers per well.

**Mouse treatment and sample collection for RNA sequencing**. Male mice between 13 and 15 weeks of age were used for the RNA sequencing experiments. The mice were first acclimated to the treadmill (Exer 3/6, Columbus Instrument, Columbus, OH) for 10 min on three consecutive days. The treadmill inclined at 10° and the speed was 0, 5 m·min$^{-1}$, and 10 m·min$^{-1}$ for day 1, 2, and 3, respectively. To minimize the effects of training on the skeletal muscles, the mice rested for 7 days before sample collection. On the day of sample collection, food was withdrawn from the mice at 9:00 AM to minimize the effects of food intake on signaling and gene transcription in the muscles. At 12:00 PM, the mice ran on a treadmill set to 10° of inclination. The treadmill speed was set at 10 m/min for 2 min to allow the mice to warm up. The speed was then increased to 15 m/min and then continuously ramped up from 15 m/min to 23 m/min at a rate of acceleration of 0.6 m·min$^{-2}$. When the running protocol ended, the mice had run 350 meters, which was lower than the average running capacity of VV mice tested by the same treadmill protocol. Any mice that did not finish the protocol were excluded from subsequent sample collection. After exercise, the mice were allowed to rest for 3 h with access to water but not food. Then they were euthanized by cervical dislocation after being anesthetized by isoflurane. The quadriceps muscles were quickly excised, frozen, and stored in liquid nitrogen.

**RNA extraction, quantification, and quality control**. To extract high-quality total RNA, the quadriceps muscles were processed first in the Trizol reagent (ThermoFisher Scientific, Catalog # 15596018) and then purified by RNeasy mini columns (Qiagen, catalog # 74104) as follows. To avoid sampling bias, the entire quadriceps muscles were homogenized in Trizol reagent at the weight (mg) to volume (µL) ratio of 1:15. One mL of homogenate was processed following the manufacture's protocol until the step of phase separation. Then, 0.5 mL of the aqueous phase was mixed with an equal volume of 70% ethanol for subsequent RNA purification by RNeasy mini kit (Qiagen, catalog # 74104) with on-column DNAse (RNase-Free DNase Set, Qiagen catalog # 79254) treatment. The concentration of the RNA was determined by Qubit fluorometric quantitation (Qubit RNA BR Assay Kit, ThermoFisher Scientific, catalog # Q10210). The integrity of the total RNA was determined by a Fragment Analyzer (Advanced Analytical Technologies, Inc). The average RNA quality number (RQN) was 9.04 ± 0.08 (Mean ± SEM, $n = 24$).

**RNA sequencing library preparation**. In total, 1 µg of the total RNA from each sample was used for RNA sequencing library preparation with the TruSeq® Stranded mRNA Library Prep kit (Illumina, catalog # 20020594). The libraries were barcoded by TruSeq® RNA Unique Dual Indexes (Illumina, catalog # 20022371) and quantified by qPCR on a Bio-Rad CFX Connect Real-Time PCR detection system with the NEBNext Library Quant Kit for Illumina (New England Biolabs, catalog # E7630L). The libraries were normalized to 10 nM, pooled, and sequenced.

**RNA sequencing and data analyses**. RNA sequencing was carried out at Johns Hopkins School of Medicine Genetic Resources Core Facility on a NovaSeq 6000 sequencing system (Illumina) with a S1 flow cell for 200 cycles, generating 100 bp paired-end reads. Sequencing data processing was carried out by the Johns Hopkins Computational Biology Consulting Core. The statistics for sequencing data analysis are presented in Tables S2–4. The RNA sequencing data were submitted to the GEO repository and were assigned record number GSE132520 (https://www.ncbi.nlm.nih.gov/geo/query/acc.cgi?acc=GSE132520).

Biological interpretation of sequencing results was carried out with the Ingenuity Pathway Analysis (IPA) platform (QIAGEN Ingenuity Systems, Redwood CA, USA) at the Johns Hopkins Deep Sequencing & Microarray Core Facility. The differentially expressed genes (over 2σ) between exercised and sedentary groups within the same genotypes and between the MM and VV samples after exercise were mapped by IPA to known pathways and biological functions of a curated Knowledge Base (QIAGEN Bioinformatics IPA Winter Release 2018). IPA evaluated the statistical significance of over-representation of the differentially expressed genes in each pathway or biological function, and a $P$ value < 0.05 was defined as statistically significant. In addition, IPA calculated whether a pathway or biological function can be considered activated ($z \geq 2.0$), inactivated ($z \leq -2.0$), or bidirectional altered ($-2.0 < z < 2.0$) based on the up- and downregulation of genes involved in the pathway or biological function.

**Mast cell preparation and treatment**. To prepare bone marrow-derived mast cells, hematopoietic progenitor cells from littermate MMδ (WT) and VVδ mice were isolated from the femur and tibia of 6–8-weeks-old mice. These mice were generated and described by us[13]. Cells were cultured at a starting density of $1 \times 10^6$ cells/mL in the presence of 10 ng/ml mouse recombinant IL-3 (BioLenged, 575908) for 4 weeks as previously described[16]. Mast cell phenotype and purity were confirmed by Toluidine blue staining and flow cytometry analysis with antibodies specific for c-Kit (APC, 17-1171-81, eBioscience/ThermoFisher, 1:200) and FcεRI (FITC, 134305, Biolegend, 1:100). For mast cell degranulation, $1 \times 10^6$ BMMCs were sensitized with 1 µg/mL anti-OVA IgE (clone E-C1, Chondrex Inc) overnight at 37 °C. Cells were washed, resuspended in Tyrode's buffer, and challenged with 10 µg/ml OVA (Sigma-Aldrich, A5503) at 37 °C. Thirty minutes after the onset of challenge, aliquots of the cells were obtained for assessment of the mast cell activation marker LAMP1/CD107a (PE, 12-1071-81, eBioscience/ThermoFisher) on a BD Accuri C6 Plus flow cytometer. The remaining cells were collected and stored in QIAzol Lysing Reagent (Qiagen) for RNA extraction 4 h after OVA treatment.

**RT-qPCR**. For samples from mice, we converted 1 µg of the total RNA from each sample into cDNA with the iScript™ Reverse Transcription Supermix (Bio-Rad, Hercules, CA catalog # 1708840). 2 ng of cDNA were used in each qPCR reaction on a CFX Connect Real-time PCR detection system (Bio-Rad, Hercules, CA) with SsoAdvanced™ Universal SYBR® Green Supermix (Bio-Rad, Hercules, CA catalog # 1725271). The primers for *CaMK2b* (qMmuCID0021273), *CaMK2d* (qMmuCIP0030149), and *CaMK2g* (qMmuCIP0030022) were pre-validated primePCR primers from Bio-Rad. For *Camk2a*, commercial primers cannot differentiate between transcripts encoding CaMKIIα and the highly abundant transcripts encoding the non-kinase adaptor protein αKAP[75]. For this reason, the *Camk2a* primers (AGGTGTGTGAAGGTGCTGG, and TGGAGTCGGACGATATTGGG) were designed with the NCBI primer design tool (https://www.ncbi.nlm.nih.gov/tools/primer-blast/) and validated in house to recognize all splicing variants

containing the kinase domain. qPCR data were analyzed by the software Bio-Rad CFX Manager 3.1, using *Gapdh* expression as the loading control.

For mast cells, 2.5 µg of the total RNA from each sample were converted into cDNA with the iScript™ Reverse Transcription Supermix (Bio-Rad, Hercules, CA catalog # 1708840). In all, 10 ng of cDNA were used in each qPCR reaction in custom-designed PrimePCR assay plates containing all target genes listed in Supplementary Data 1 and three additional reference genes (*Hprt*, *Rps18*, and *Ppia*) on a CFX384 Real-time PCR detection system (Bio-Rad, Hercules, CA). The primer validation data for primers on the assay plates are provided by Bio-Rad and are included in Supplementary Data 3. qPCR data were analyzed by the software Bio-Rad CFX Manager 3.1. Three target genes failed to be amplified, leaving 87 genes for analysis as shown in the Source Data for Fig. 4b.

For flies, the total RNA from adults was prepared with Direct-Zol RNA miniprep kit with on-column DNase treatment (ZYMO Research, #2050). cDNA synthesis and qPCR were carried out as described above. The primers for qPCR were obtained from the FlyPrimerBank[76] and were further validated for efficiency and specificity. Expression of RP49 was used for normalization. The primer IDs and sequences are as follows: RP49 (PD41810: AGCATACAGGCCCAAGATCG, TGTTGTCGATACCCTTGGGC), THOR (PD43730: CAGATGCCCGAGGTGT ACTC, CATGAAAGCCCGCTCGTAGA), SULFIREDOXIN/CG6762 (PD42226: GCATCGATGAGACCCACCTG, GATCCACAGCAGGTCGATGG), GADD45 (PD42384: GGCCTTTTGCTACGAGAACG, CGCAGTAGTCGACTAGCTGG), SOCS36E (PP11279: ATGGGTCATCACCTTAGCAAGT, TCCAGGCTGATCGT CTCTACT), GSTD2 (PP27238: AAACCGCGTTTGGATTTCTCG, GTGGAGAC AGTGGACAGGAT), ANK2 (PD41602: TGTGGTCATGTTAGGGTGGC, TTCA AAGCCCTTGCATTGGC), KAY (PA60087: ACTCCAACGCTTCGTACAACGA TA, CACTTGAAGTATCCGGTCGTGTC), LK6 (PA60203: CAAACGCCCAGTA ACATC, GCTGTAGGACCACACGCTTGAC), ATF3 (PP9314: AAGACGCCAG AGATCCTCAAC, GCAACTGGAATGACTGCTGTC), DREP3 (PP34108: GAC GATGGTTTGGACGATGC, TGTTCCTCGTGATGTCCTTGA), HTK (PP1052: TACCTGGTACATTACACAGGCT, GTGCGAGTTTTCTGCTTGGA), SNF4Aγ (PP10614: ACCTCCGCCAAGTTGGTTG, CGCACACCGTTGTAGACGA), SRA (PP5475: CCGATGCACCTGATCCGAC, TTGTTCTTGCTTCTGCCGTTG), AXUD1 (PP35701: GAGATAATCGTACTAGGCGATGC, GCGGAGTCAA GAATGTTGTCAA).

**Statistical analysis.** All statistical analyses except the RNA sequencing analyses were performed with GraphPad Prism 8.3 (GraphPad Software LLC). When applicable, we first performed normality tests (D'Agostino–Pearson). If the data passed the normality test, we applied parametrical tests without assuming equal standard deviations among groups. If the data did not pass normality tests, we performed nonparametric tests. All methods we used are detailed in figure legends, and the *P* values are also provided in the figures.

**Reporting summary.** Further information on research design is available in the Nature Research Reporting Summary linked to this article.

## Data availability
Raw RNA sequencing data and the gene expression matrix are available in the Gene Expression Omnibus (GEO) under accession number GSE132520. All other data are available within the paper and its supplementary information files. Publicly available data used in this study were retrieved as follows: Most CaMKII orthologues listed in Supplementary Fig. 1 and Supplementary Table 1 were identified in the Interpro database (http://www.ebi.ac.uk/interpro/entry/IPR013543/taxonomy). Additional sequences were uncovered by BLAST in the NCBL nucleotide database and translated into proteins (https://blast.ncbi.nlm.nih.gov/Blast.cgi). All sequence files with accession numbers are available in Supplementary data 2. Human mast cell transcriptome profiling dataset GSE125887 was retrieved from GEO. Transcriptome changes of *Drosophila melanogaster* in response to paraquat treatment were obtained from Supplementary Table 9 of the reference[54] (https://static-content.springer.com/esm/art%3A10.1038%2Fnature12962/MediaObjects/41586_2014_BFnature12962_MOESM304_ESM.zip). Source data are provided with this paper.

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

## Acknowledgements

We thank Drs. Hal Dietz and Gregg Semenza for their insightful comments and suggestions, Dr. Andrew Feinberg for sharing instruments, Teresa Ruggle for assistance in graphic design, Benjamin Garlow for assistance in developing KTR, Jinying Yang for managing mice, Tran Nguyen for maintaining fly stocks, Dr. G. William Wang for technical help in insulin sensitivity and glucose tolerance tests. This work was supported by the National Institutes of Health (R35-HL140034 to M.E.A., R37-AR055099 to E.O.H. and M.F.S., R01-AR059179 and R21-AR067872-01 to R.M.L., R01-HL124091 to M.C.V. and A.C., R01-NS079584 to M.N.W., R21-NS108842 to M.A.B.) and by the Michel Mirowski Discovery award at Johns Hopkins University to M.A.B.

## Author contributions

Q.W., G.S.B., and M.E.A. contributed to the conception and design of the work; Q.W., E.O.H., M.C.V., I.D.B., D.C.D., J.M.G., K.R.M., A.W., S.A., N.L., D.M., K.R.W., and R.M. L. contributed to the acquisition of the data; Q.W., E.O.H., M.C.V., I.D.B., K.R.M., A.W., S.A., C.M.A., L.D.F., C.C.T., D.M., S.R., M.N.W., A.C., P.G., M.A.B., and M.S.F. analyzed and interpreted the data; Q.W., G.S.B., and M.E.A. drafted the manuscript; Q.W., G.S.B., and M.E.A. substantively revised the manuscript.

## Competing interests

The authors declare no competing interests.
