## [Peer Review File · Nature Communications]

Reviewers' Comments:

Reviewer #1:

Remarks to the Author:

The manuscript by Wang et al examines the physiological roles of the redox sensitivity of CaMKII and their evolutionary implications. Vertebrate CaMKII contains an amino acid motif that makes it redox sensitive. This motif is absent in most invertebrates. The authors use two animal models: the mouse and the fruit fly. In the mouse, they show that the absence of the redox -sensing motif impairs muscle function important in exercise and impairs aspects of immune function. In contrast, other studies have shown that the presence of this motif contributes to certain age-related diseases. In the fly, they show that the introduction of redox-sensing motif improves aspects of muscle function while making flies susceptible to a redox challenge. The authors consider the evolutionary implications of these findings.

I found the experiments well preformed, the data well analyzed and in most cases suitably interpreted. I do think however that there are certain aspects of the manuscript that would have to be improved before I could recommend its publication in Nature Communications.

My assessment is mostly focused on the fly work and how it relates to the mouse, as I do not have expertise in mouse muscle or mouse exercise/immune physiology.

Main comments:

1) I find the framing of the work in the context of antagonistic pleiotropy, the evolutionary theory explaining the existence of aging, quite awkward and sometimes forced. I should note that I find the hard sell on antagonistic pleiotropy is not required for this paper. I could be softened.

1A) Firstly, I think the authors confuse antagonistic pleiotropy, a theory that tries to explain the existence of aging, for a theory that explains the mechanism(s) of aging.

1B) There is also a misunderstanding of the theory: page 2 lines 8-9: the early beneficial traits are favored not because of the late-acting detrimental effects but despite them.

1C) The authors imply there is not much evidence for antagonistic pleiotropy. However, I think there are quite a few examples where the known gene function fits the expectations: insulin/IGF pathway, the TOR pathway; similar for processes such as protein synthesis, cellular senescence etc.

1D) In the fly, the authors show that CaMKII being made redox sensitive provides apparent advantages in youth (specifically muscle performance) at expense of oxidant resistance. But this is not exactly antagonistic pleiotropy as it pertains to aging. Oxidant sensitivity is not the same as aging. A lifespan experiment would be more persuasive. The case is much stronger for mice where the absence of redox sensitivity ameliorates age-related diseases.

2) I think a more detailed characterization of the VV and MM flies would be helpful.

2A) For example, are the two the same size as smaller flies often climb faster? Muscle function can be linked to metabolic phenotypes – what are their TAG stores, starvation sensitivity, feeding frequency like? Do they have the same rate of development, same fecundity? I think this would help understand how specific the described phenotype is and in turn reassure the reader that the interpretation given by the authors is likely to be correct.

2B) The interpretation of the fly data hinges on the idea that the MM protein is more susceptible to oxidation. However, this may not be the case (for example if the MMs are not solvent accessible in the fly protein). Could the authors directly measure this? Additionally, how can the authors be sure that the MM mutation is not disrupting normal CaMKII function in the fly? Have they compared the phenotypes of MM flies to those with CaMKII loss of function? Maybe the MM mutant is simply a loss of function or a partial loss of function in the fly.

2C) I think the initial presentation of the fly model in 1st section but then further characterization of the model later doesn't help (I think a different order may help with understanding of the data).

Minor:

Page 3 lines 19-22: I don't understand this – please clarify.

CM and MM are treated the same – are they really the same with respect to redox sensitivity?

Reviewer #2:

Remarks to the Author:

The Wang et al. manuscript entitled "A Critical Performance/Disease Trade-off at the Dawn of Vertebrate Evolution" asserts that CaMKII represents one of the best examples of antagonistic pleiotropy, in which aging related sequences are permitted because they confer early life advantages. The initial evidence for this is that the CaMKII gene family in vertebrates contains an extremely well conserved pair of Methionines (MM) or a Cysteine/ Methionine pair (CM) that are the target of harmful oxidation by reactive oxygen (ROS). These sequences (MM/CM) are strikingly absent from invertebrates and are often replaced with nonpolar residues such as V (Fig S1). This is a truly noteworthy observation and there is an opportunity to directly test the central hypothesis by swapping MM for VV and vice versa depending on the host species. The concept that the MM/CM pair confers physiological advantages at young ages, even though their oxidation has been associated with aging-related traits can therefore be tested. Support for the latter studies is extensive and cited in this manuscript to generally show that mutation of the MM/CM to VV confers resistance to aging-related diseases.

Using a nicely complementary approach, CaMKII in flies and mice are mutated. The single *Drosophila* CaMKII sequence (VV) was mutated to MM and homozygous MM/MM flies were then evaluated. In mice, the gamma CaMKII gene (*camk2g*) was mutated from MM to VV and homozygous VV mice evaluated.

The direct tests of the central hypothesis come from functional experiments in the genetically engineered mice and flies that are then subjected to assays for exercise and immunity and sensitivity to an ROS-generating toxin, paraquat, followed by transcriptional profiling. Mutant mice were similarly healthy to wild type mice. The mouse exercise assays included speed and persistence in an induced running model. The differences between these mice are small but are statistically significant (Fig 2B,C). The immunological assays were mast cell degranulation.

This study is original and would be novel and important to the field. However, there are two significant omissions in this study that should be addressed.

1. The first significant concern is that a direct assessment of endogenous CaMKII oxidation is never conducted.

It is justifiably asserted that when the single CaMKII gene in *Drosophila* is mutated from VV to MM, the mutant flies are now different from wild type flies because CaMKII can be oxidized in the mutants on the MM residues. The author's laboratory itself has shown (Erickson, 2008) that oxidized endogenous CaMKII can be detected in heart lysates using an antibody generated in their lab, but such immunoassays were never used in this study. There are sufficient sequence differences that might preclude cross-reactivity in flies using this antibody, so this may explain why this assay was not used in *Drosophila*, but it is not mentioned and should be.

In a complementary fashion, VV mice would be predicted to have diminished endogenous CaMKII oxidation in comparison to wild type mice either in skeletal muscle or in cardiac tissue as previously demonstrated (Erickson, 2008). This could be conducted in comparison to WT mice after exercise to demonstrate that this mutant is resistant to oxidation at this site. The mouse transcriptional profiles are conducted in isolated quadriceps muscle, so lysates from this tissue could also be assessed for ox-CaMKII to show more tissue specificity.

If this antibody is no longer available, another potential way in which CaMKII could be directly

assessed is through CaMKII activity assays. Oxidized CaMKII should have enhanced Ca/CaM independent activity. CaMKII activity assays are sensitive and quantitative and can be conducted on lysates of tissue in wild type and VV mice.

Another analysis that could have benefitted from anti-oxCaMKII antibody was in the fatigued mice (Fig 2H-J) including mice with the Ncf1-background. Ncf1 encodes a co-factor for NADPH oxidase, which is one enzymatic source of ROS. Thus, these mutants would be predicted to have diminished ox-CaMKII.

The "KTR" reporter was developed and then used in mice to evaluate the ability of endogenous wild type and mutant CaMKII to influence nuclear to cytoplasmic localization changes. In supplemental figure S5, this reporter clearly translocates from the nucleus to cytoplasm in cultured cells (RPE-1) after histamine treatment and is blocked by a kinase dead CaMKII or a CaMKII inhibitor and therefore works nicely as described.

However, if it is intended to be used to assess altered CaMKII function, it is only used in a limited number of experiments (Figure 1D), the mouse myotube examples are not convincing and the n values are small, with two data points (of the 11 total) responsible for the majority of the difference. Does KTR translocate in flies?

2. The second omission is that lifespans are not reported. It would be predicted that mutating mice and flies would alter their lifespans. In fact, the mouse knockout mutation of the MsrA gene, which encodes a reductase that can reverse the oxidation of CaMKII residues, results in extended lifespans (Ruan 2002). It is therefore predicted that VV mice would have extended lifespans and MM flies would have reduced lifespans, but neither of these were assessed for lifespan differences. Ruan H, et al (2002) PNAS (USA) 99: 2748.

Other concerns:

1. In mice, gamma CaMKII is transcriptionally assessed by qPCR and shows no difference between wild type and mutant, but none of the other CaMKII genes are evaluated in mutants. This is in spite of data in the same figure (S4) showing the qPCR evaluation of all 4 mouse CaMKII genes, so primers are available. There is strong evidence in vertebrates that germ line mutation knockout of one CaMKII gene does not necessarily lead to the predicted change in CaMKII activity (Bucks (2010) PNAS 107:81; Gagnon (2014) PLOS One 9: 98186; Rothschild (2020) 742:144567).
2. Transcriptional responses are poorly interpreted and don't make a compelling point ("the up-or down regulation of a smaller number of genes was completely blunted or even reversed.") Fig 3D.
3. Can ROS be measured in paraquat samples? This would also be useful confirmatory information.

Reviewer #3:

Remarks to the Author:

In Wang et al, the authors look at potential antagonistic pleiotropic effects of the CAMKII protein. They find a specific two amino acid motif is fixed in vertebrates while a separate two amino acid motif is almost completely fixed in invertebrates. The vertebrate version of the protein improves skeletal muscle function in response to exercise. I found the paper interesting, and the authors appear to have done a fairly thorough job in their experimental design and execution on the effects of the CAMKII amino acid substitutions. I think this paper has the potential to be of interest to both molecular and evolutionary biologists, and I found the methods overall strong though I think a little more detail is needed to make them completely reproducible. I have listed several issues and general questions below. I also understand that with COVID, completing some of the follow-up experiments might not be possible.

-With regards to the interpretation that this is an "antagonistic pleiotropy" protein, I think the evidence is lacking. While improved muscle performance in flies and mice with the MM motif are shown, the negative effects of the gene are not thoroughly investigated. While the authors show the

MM flies are less resistant to oxidative stress, this is not necessarily a good measure of overall fitness, especially as the oxidative theory of aging is not really supported any longer. While paraquat is often used to induce ROS in flies, this is a super, toxic stressor and is not necessarily similar to the natural increase in ROS with age seen in flies. In addition, if this increase in sensitivity to ROS is the reason invertebrates have fixed the VV sequence, why do vertebrates not also show this sensitivity and thus negative fitness that you are arguing? It would have been helpful to show a) the effects of oxidative stress on your mouse models as well as b) the effects of the MM vs VV flies on reproduction. Do the MM flies reproduce less? Have slower development? Or some other more commonly associated fitness measurement. Similarly, do the MM mice show any difference in reproduction compared to VV mice? While the authors do show that reducing ROS in the muscles of the mice reduces their performance, I think it would be better if they showed what increased ROS did the mice, as that was experiment done in the flies. If the authors can show these, I think they will have a much stronger argument for antagonistic pleiotropy at this locus.

-It appears you only used male mice while the majority of your fly experiments were done in females. Is there a reason you did not use both sexes in your study, as stronger antagonistic pleiotropic effects could be sex specific?

-I would like to see more detail on the description of animal husbandry for both the flies and mice. Temperature? Humidity? Diet? Were the animals fed ad libitum?

-Is there a reason you only measured mortality at 24 and 48 hours in the paraquat flies? It would be standard to determine number dead at 8- or 12-hour intervals and calculate actual survival curves.

-You state they are in vials of 10 but how many replicate vials were used? I assume 10 replicates based on the sample sizes in the figure legends, but I would state clearly in the methods

-Please provide a statistical analysis section in the methods. In some of the figure legends you mention the stats used, but in others you do not. It would be much clearer if you would just state all the analyses used in the methods. In addition, did you make sure that your data met the assumptions of the different statistical tests used?

-Looking at Table S1, it appears that the only vertebrates without the MM/CM module are within the ray finned fishes. Any thoughts as to why this group might have lost this specific module? In addition, are the 27 fish species without the module all sister species, such that the loss occurred once on the phylogeny?

-On page 4 line 6, I believe you have the WT next to the wrong genotype.

-Along the same lines, I found the constant reference to WT to be confusing. I think you clearly laid out that MM is the WT in mice while VV is the WT in flies, and I would remove the constant (WT) reference.

-Please include the ages of the animals used in experiment. You state 13-15 weeks in the RNAseq analysis, was this the age for all the other mouse experiments as well?

-Your mice appear to only be around 10% body fat (Figure S6), this seems really low compared to most B6 mouse studies, as well as most other strains. Is this due to young age or potentially a husbandry effect?

-Page 4 line 11, you state the reduced exercise performance in VV mice could be due to "metabolism or motivation". Did you measure metabolism or behavior in these animals? While measuring motivation is difficult, if not impossible, other basic behavioral measures, besides voluntary wheel running, might provide some insights into the differences with these two genotypes.

-The *ncf1* mice are not listed in your methods at all. Please include where these came from, ages, sex studied, etc.

-Page 2 line 10 and page 3 line 25 use the word "enjoy". However, this word is not correct for evolutionary derived characters, please change.

-I found it interesting that you found differences in inflammation between the MM and VV mice. As low grade inflammation is now thought to be a potentially large contributor to aging, have you aged out these mice to see if the VV mice age differently than the MM mice? If MM tends to promote inflammation, potentially the VV mice are longer lived? This is most likely its own study, but I think it might be nice for the authors to hypothesize how this inflammatory response might effect aging in the discussion. Along the same lines, are their differences in longevity in the VV and MM flies? This would be a much simpler/quicker experiment.

Reviewer #4:

Remarks to the Author:

Wang et al study the interesting case of CaMKII, a signaling molecule with a pleiotropic effects. Specifically, they analyze how the vertebrate specific acquisition of an (Reactive) oxygen-sensing domain led to outstanding benefits in muscle performance, but led to the increased risk of cardiac failure in late life. Hence, CaMKII would be represent an exquisite candidate to fit the predictions of the antagonistic pleiotropism theory of aging, proposed by George Williams. The manuscript is well written and presents a large amount of compelling results. However, there are some aspects that need careful consideration. The main assumption of the manuscript is that the radical theory of aging is correct and ROS are bad. However, the impact of ROS on aging has been completely reconsidered over the past

years in light of the theories of hormesis (e.g. mitohormesis). ROS would trigger "rescue" modes in organisms, which would benefit healthspan. This should be discussed and the results presented should be framed also in light of hormesis. Conceptually, in the introduction there is a slight misconception in that all the genetic variants present in nature are there because of the action of natural selection. However, there is a lot of genetic

variation that has nothing to do with strong purifying selection (lines 28-29 of page 2). Also, the authors did not mention the mutation antagonistic (MA) theory of aging. We encourage them to cite it and to speculate whether their observations would fit MA, rather than AP.

I would find very interesting to see multiple alignments for CaMKII across different mammals. This should be pretty straightforward to do. The authors could show how conserved this domain is in species with different degrees of muscle performance and thermal requirements. How conserved is the MM/MC domain across mammals? The strong amino acid conservation of the CM/MM module is indicative for a strong purifying selection and could be preceded by initial positive selection in early vertebrates leading to fixation. Although the positive selection signal might be obscured by the rapid fixation and consequent purifying selection since stem vertebrates, measuring the strength of selection using Dn/Ds for the CaMKII genes would strengthen the argument. The transcription response upon 10mM paraquat shown in Figure 4 is significantly higher in MM vs. VV genotypes, but overall the direction of response is very similar between MM and VV. The ROS-signaling MM genotype seems to induce a stronger response of the same transcriptional machineries. Could the dose dependent response be responsible for the detrimental effects of the MM genotype late in life? The most critical aspect of the whole paper is that indeed the CaMKII MM domain is selected for by natural selection (it improves fitness) but it remains harmful late in life, representing a risk for aging/death. The authors refer to extensive literature in this respect, connecting CaMKII function/activation during stroke and cardiac failure. However, they do not show (or maybe I missed it?) this in their own experimental work in mice. I could not see evidence that VV is protective over MM in terms of (for instance) stroke risk or cardiac failure. In the parts where it should demonstrate AP, the study

becomes needlessly complicated, yet fails to provide sufficient evidence. It has shown that MM mice outperform VV mice; but do VV outperform MM mice in some viability metric (or demonstrate a deceleration of the aging phenotype) under usual circumstances. If VV is protective to aging-related damage in mice, based on the authors own work, I would be very impressed and be convinced by their argument.

Fly work: The adverse effects of the CM/MM module shown in flies are very strong. My concern is that the levels of paraquat used in the experiments are not comparable to normal physiological levels. Can the author comment on this and if necessary, clarify this in the manuscript. The MM genotype in flies seems to perform better in climbing, shortening velocity and relaxation rate in flies (Figure 6). This effect is diminished when adding paraquat, a ROS inducing toxin. The fact that there is an advantage for the MM

genotype in flies, but it was not fixed in the invertebrate lineage would implicate that invertebrates maybe encountered higher level of ROS in their environment? How is the performance of climbing in mice when exposed to higher ROS levels? Is the concluded trade-off connected to a benefit in a feature that is missing in invertebrates like skeletal muscle or maybe even adaptive immunity?

There are some typos and parts that are unclear:

- page 4, line 6: MM and WT(VV). It should be MM(WT) and VV.
- page 4, line 13: fig. 7Aa does not exist. Is it perhaps fig. S7A?
- Figure 3: what is the difference between E and F? I could not grasp it from the figure and from its caption.
- Figure 3 should also show what's up-regulated in VV vs. MM.

Response to Reviewers/summary comments

We thank the reviewers for their time and constructive comments. We performed a series of new experiments in response to the critiques and have substantively revised the manuscript.

Importantly, our new studies show that unperturbed MM flies (harboring a redox sensitive knockin mutation derived from vertebrates) have shorter lifespans than wild type VV flies (see revised Fig. 1d, e and Supplementary Fig. 6). The MM mutation confers oxidative activation of CaMKII, as we have shown previously, and demonstrated in this publication using a recently validated CaMKII activity reporter¹ in living cells (see revised Fig. 1f-i). The shortened lifespan of MM flies, taken together with oxidation induced frailty (Fig. 6a, b), organ dysfunction (Fig. 6c-e) and death (Fig. 1c-e and Supplementary Fig. 6) and the early life physiological enhancement by MM CaMKII (in flies (Fig. 5) and mice (Fig. 2a-j, 3b-h, 4b)) fulfills the precepts supporting the theory of antagonistic pleiotropy. We also provided an expanded description and discussion of the evolutionary tenets of this theory and the predictions that it makes; we hope the referees agree that these changes improve the clarity of the involved ideas. In our original manuscript we had not demonstrated an effect of MM CaMKII on lifespan, in the absence of paraquat induced oxidative stress. Our new findings are remarkable and convincing and we hope will satisfy the request to address this aspect of antagonistic pleiotropy theory. We originally considered a longevity study in mice. However, in addition to the prohibitive time-line of such an experiment, mice (and humans) have 4 CaMKII genes residing on 4 separate chromosomes. While it is, in theory, possible to imagine a global MM to VV knockin mouse where all 4 CaMKII isoforms are mutated, this would require a very long and expensive process, including making new knockin mice for CaMKII α and CaMKII β (we have CaMKII δ and CaMKII γ knockins) and an elaborate and time consuming breeding protocol that would take many years to accomplish and is beyond the scope of this study. In contrast, the fly, with a single CaMKII gene and a relatively short life span, is ideal for this experiment.

We also performed new studies to measure fecundity (Supplementary Fig. 4), oxidative response to paraquat (Supplementary Fig. 5), size of MM and VV flies (Supplementary Fig. 16a), and metabolism (Supplementary Fig. 16b). We increased the number of studies in isolated, field stimulated, skeletal muscle cells expressing the CaMKII activity reporter (Fig. 1g-i). Importantly, these skeletal muscle studies confirmed the original work – demonstrating that MM is essential for CaMKII activation by oxidation in electrically stimulated living skeletal muscle cells. As requested, we performed new studies in isolated skeletal muscle from mice lacking p47 (*Ncf1*^{-/-}), an important component of NADPH oxidase in muscle; the *Ncf1*^{-/-} fibers showed markedly reduced CaMKII activation during electrical stimulation, supporting our original studies using the antioxidant N-acetyl cysteine (NAC) that similarly eliminated CaMKII activation by electrical stimulation. We also replaced the original confocal images for KTR experiments with the FDB fibers with more representative images (Fig. 1f and Supplementary Fig. 10). We used structural and computational modeling that revealed the highly homologous fly and mammalian CaMKII regulatory domains are predicted to display the MM residues in a manner where they are accessible to oxidation in the aqueous phase (Supplementary Fig. 3).

A detailed series of responses follow. Important changes to the revised manuscript are marked by yellow highlighting.

Reviewer #1 (Remarks to the Author):

The manuscript by Wang et al examines the physiological roles of the redox sensitivity of CaMKII and their evolutionary implications. Vertebrate CaMKII contains an amino acid motif that makes it redox sensitive. This motif is absent in most invertebrates. The authors use two animal models: the mouse and the fruit fly. In the mouse, they show that the absence of the redox-sensing motif impairs muscle function important in exercise and impairs aspects of immune function. In contrast, other studies have shown that the presence of this motif contributes to certain age-related diseases. In the fly, they show that the introduction of redox-sensing motif improves aspects of muscle function while making flies susceptible to a redox challenge. The authors consider the evolutionary implications of these findings.

I found the experiments well preformed, the data well analyzed and in most cases suitably interpreted. I do think however that there are certain aspects of the manuscript that would have to be improved before I could recommend its publication in Nature Communications.

Thank you for these positive comments.

My assessment is mostly focused on the fly work and how it relates to the mouse, as I do not have expertise in mouse muscle or mouse exercise/immune physiology.

Main comments:

1) I find the framing of the work in the context of antagonistic pleiotropy, the evolutionary theory explaining the existence of aging, quite awkward and sometimes forced. I should note that I find the hard sell on antagonistic pleiotropy is not required for this paper. I could be softened.

Thank you for these comments. Upon reflection, we agree that our original data did not fully support a “hard sell” for antagonistic pleiotropy. However, our new data substantially fortify this premise, and meet the predictions of antagonistic pleiotropy. In particular, our new data (Fig. 1d, e and Supplementary Fig. 6) show that the MM mutation shortened lifespan in unperturbed male and female flies at 25°C and 29°C, strengthening the case that MM CaMKII is, indeed, a previously unappreciated example of antagonistic pleiotropy.

1A) Firstly, I think the authors confuse antagonistic pleiotropy, a theory that tries to explain the existence of aging, for a theory that explains the mechanism(s) of aging.

Thank you for this comment. Indeed, we agree that our study is focused on a mechanism that could underlie the theory of antagonistic pleiotropy rather than on the theory itself. However, the theory does make predictions that help explain our data.

1B) There is also a misunderstanding of the theory: page 2 lines 8-9: the early beneficial traits are favored not because of the late-acting detrimental effects but despite them.

Thank you. We have revised this language and the tenets of the theory are conveyed more clearly.

1C) The authors imply there is not much evidence for antagonistic pleiotropy. However, I think there are quite a few examples where the known gene function fits the expectations: insulin/IGF pathway, the TOR pathway; similar for processes such as protein synthesis, cellular senescence etc.

Thank you for these comments. We have added a reference (reference 6) and expanded the text to make this point (see page 2).

1D) In the fly, the authors show that CaMKII being made redox sensitive provides apparent advantages in youth (specifically muscle performance) at expense of oxidant resistance. But this is not exactly antagonistic pleiotropy as it pertains to aging. Oxidant sensitivity is not the same as aging. A lifespan experiment would be more persuasive. The case is much stronger for mice where the absence of redox sensitivity ameliorates age-related diseases.

Thank you for these points. We performed the requested lifespan experiments with flies. We tested male and female flies at 25 °C and 29 °C and found that MM flies (both males and females) have shorter lifespan under both temperature conditions (Fig. 1d, e and Supplementary Fig. 6). Our new results thus revealed a striking effect for ROS-sensitive CaMKII to promote aging. We believe these new results better support our view that the MM module in CaMKII provides a new example and molecular mechanism for how a gene, satisfying the precepts of antagonistic pleiotropy, can contribute to aging.

2) I think a more detailed characterization of the VV and MM flies would be helpful.

We performed more detailed characterization of the VV and MM flies (see below).

2A) For example, are the two the same size as smaller flies often climb faster? Muscle function can be linked to metabolic phenotypes – what are their TAG stores, starvation sensitivity, feeding frequency like? Do they have the same rate of development, same fecundity? I think this would help understand how specific the described phenotype is and in turn reassure the reader that the interpretation given by the authors is likely to be correct.

We thank the reviewer for asking for a more detailed characterization of the flies to provide better context to understand the phenotypes we reported in the manuscript. Because the climbing performance was one of the major phenotypes we compared between MM and VV flies, and the reviewer pointed out that the size of flies may affect climbing performance, we

measured the size of the female flies at 5 days of age; this is the same age as the flies we used to test climbing performance. We found no difference in body size between the genotypes (Supplementary Fig. 16a), suggesting that body size difference did not contribute to climbing performance difference. We also assessed TAG stores in the same groups of flies, as requested, because muscle function can be linked to metabolic phenotype, as suggested by the reviewer. We found that the MM flies have a significantly smaller TAG store (Supplementary Fig. 16b). This is interesting because CaMKII was shown to promote lipolysis, which in turn may affect exercise performance. We revised the manuscript (page 8) and added relevant references (55, 56) related to this new finding.

We also performed a lifetime fecundity assay (Supplementary Fig. 4). We recorded the progeny produced by single females in consecutive 6-day windows throughout their lifespan, and we found that MM and VV flies reproduce similarly through their lifespan (Supplementary Fig. 4a), which resulted in similar total numbers of progeny per female per lifetime (Supplementary Fig. 4b). We also observed that both MM and VV flies take 10 days to develop from eggs to adults at 25 °C, suggesting no significant difference in developmental rate. We think that these negative results and the *in situ* heart tube experiments that minimized the variables from feeding and environmental conditions have sufficiently added to our confidence that an increase in intrinsic muscle performance underlies the climbing and cardiac contractility phenotype. Therefore, we did not perform additional metabolic experiments such as starvation sensitivity and feeding frequency due to the limited lab time available under the COVID-19 situation.

2B) The interpretation of the fly data hinges on the idea that the MM protein is more susceptible to oxidation. However, this may not be the case (for example if the MMs are not solvent accessible in the fly protein). Could the authors directly measure this? Additionally, how can the authors be sure that the MM mutation is not disrupting normal CaMKII function in the fly? Have they compared the phenotypes of MM flies to those with CaMKII loss of function? Maybe the MM mutant is simply a loss of function or a partial loss of function in the fly.

Thank you for these insightful questions. We first assessed the solvent accessibility of wildtype and MM mutant fly CaMKII using molecular and computational modeling, based on the human CaMKII α structure (Supplementary Fig. 3). We found that, similar to the human CaMKII, the VV residues and their replacement MM residues (located strategically in the regulatory domain) are exposed to solvent, and therefore susceptible to oxidation from the aqueous phase. Our modeling also showed that replacing the VV residues in fly CaMKII to MM does not alter the structure of the CaMKII regulatory or catalytic domains. Therefore, the VV to MM mutation likely preserves normal CaMKII function, consistent with our original report using CaMKII δ^2 . Our *in vivo* data also support this model. When we minimized ROS in the climbing experiment by treating both MM and VV flies with NAC (N-Acetyl Cysteine), the performance of MM flies regressed to that of VV flies, while the performance of VV flies was unaffected by NAC (Fig 5). We interpret these findings to suggest MM- and VV-CaMKII have equivalent activity in the absence of oxidation. The observations that MM flies gain performance advantages in the

presence of physiological ROS, and are impaired in the presence of excessive ROS (paraquat treatment) further support the concept that MM-CaMKII in flies indeed transduces signals through oxidation.

We initially considered, but rejected, the idea that the MM mutation could, unexpectedly, be contributing to a loss of function phenotype. First, our KTR (CaMKII activity reporter, recently validated and published¹ – Mesubi, JCI (2021) Fig. 4) measurements showed MM-CaMKII has more, not less, activity than VV-CaMKII (Fig. 1g-i). Second, the relatively greater susceptibility of MM compared to VV flies to oxidant injury was more consistent with CaMKII hyperactivity. Finally, we recently reported³ (Konstantinidis, JCI (2020) Fig. 5) a different line of flies harboring a hypomorphic CaMKII mutation. We made a knockin replacement of an evolutionarily ancient methionine (M308V) in the calmodulin binding domain. M308 is present in CaMKII of invertebrates and vertebrates. Flies with the hypomorphic M308V CaMKII show reduced activity (Konstantinidis, JCI (2020) Fig. 5A-D) and exhibit a striking reduction in heart tube contractility (Konstantinidis, JCI (2020) Fig. 5E-J)³, the opposite of the phenotype we observed in MM flies. Taken together, these findings provide strong support that the MM module augments CaMKII activity in the presence of ROS. Based on your comments, we revised the manuscript to make these points (page 11).

2C) I think the initial presentation of the fly model in 1st section but then further characterization of the model later doesn't help (I think a different order may help with understanding of the data).

Thank you for this suggestion. We carefully weighed our presentation versus the alternative presentations of either exhibiting all fly data upfront or pushing all fly data to the end of the paper. We respectfully ask to keep the general order of our presentation based on the following reasons.

During the revision, we have made extensive changes to the Introduction and beginning of the Results, which makes the overall logic of the manuscript clearer. Our presentation starts with the generation and initial characterization of MM flies for paraquat sensitivity (and aging in the revised manuscript). We used these fly data to support the concept that gaining the MM module was sufficient to convert the ancestral CaMKII into a ROS sensitivity enzyme. Furthermore, these fly data together with previously published mouse data also show that the detrimental effect of CaMKII oxidation is indeed an evolutionarily conserved property that likely existed when MM/CM-CaMKII first evolved in the ancestral vertebrates. This ancient origin of the detrimental effects of CaMKII oxidation allowed us to hypothesize a candidate physiological function of CaMKII oxidation: to "allowed ROS to enhance skeletal muscle performance" based on the fact that "the vertebrate stem lineage was witness to a major shift in behavioral ecology (from sessile filter feeders to active predators)." Then we tested this hypothesis by shifting our attention to mice. After confirming that CaMKII oxidation can improve muscle function and

regulate gene expression, we tested these roles in flies. Although our data revealed a striking effect for ox-CaMKII to confer a performance advantage in flies, this effect is not a priori evident for flies because evolution did not fix ox-CaMKII in flies. Because of this, we think it would be contrived to test the effects of the artificially introduced MM module on motor function and gene expression in flies at the beginning of the paper, before the mouse experiments.

On the other hand, if we pushed all fly data to the end of the manuscript, the following argument in the paper couldn't be made before we characterize the mice: "given that this conserved pattern (of the detrimental roles of CaMKII oxidation) extends across the entire breadth of the vertebrate crown clade, whatever fitness benefits are responsible for its underlying positive selection must have been in place before the origin of this crown and thus present along some length of the vertebrate stem lineage. It follows that these benefits are likely related to an emerging evolutionary scenario that set these vertebrate ancestors apart from their invertebrate relatives."

Minor:

Page 3 lines 19-22: I don't understand this – please clarify.

We revised, expanded, and hopefully clarified this text.

CM and MM are treated the same – are they really the same with respect to redox sensitivity?

In our original publication² (Erickson Cell 2008) describing CaMKII oxidation as an important mechanism for activation, we showed that oxidation of CM (as in CaMKII α) and MM (as in the other 3 CaMKII isoforms) similarly contribute to CaMKII activation.

Reviewer #2 (Remarks to the Author):

The Wang et al. manuscript entitled “A Critical Performance/Disease Trade-off at the Dawn of Vertebrate Evolution” asserts that CaMKII represents one of the best examples of antagonistic pleiotropy, in which aging related sequences are permitted because they confer early life advantages. The initial evidence for this is that the CaMKII gene family in vertebrates contains an extremely well conserved pair of Methionines (MM) or a Cysteine/ Methionine pair (CM) that are the target of harmful oxidation by reactive oxygen (ROS). These sequences (MM/CM) are strikingly absent from invertebrates and are often replaced with nonpolar residues such as V (Fig S1). This is a truly noteworthy observation and there is an opportunity to directly test the central hypothesis by swapping MM for VV and vice versa depending on the host species. The concept that the MM/CM pair confers physiological advantages at young ages, even though their oxidation has been associated with aging-related traits can therefore be tested. Support for the

latter studies is extensive and cited in this manuscript to generally show that mutation of the MM/CM to VV confers resistance to aging-related diseases.

Using a nicely complementary approach, CaMKII in flies and mice are mutated. The single Drosophila CaMKII sequence (VV) was mutated to MM and homozygous MM/MM flies were then evaluated. In mice, the gamma CaMKII gene (camk2g) was mutated from MM to VV and homozygous VV mice evaluated.

The direct tests of the central hypothesis come from functional experiments in the genetically engineered mice and flies that are then subjected to assays for exercise and immunity and sensitivity to an ROS-generating toxin, paraquat, followed by transcriptional profiling. Mutant mice were similarly healthy to wild type mice. The mouse exercise assays included speed and persistence in an induced running model. The differences between these mice are small but are statistically significant (Fig 2B,C). The immunological assays were mast cell degranulation.

This study is original and would be novel and important to the field. However, there are two significant omissions in this study that should be addressed.

Thank you for your review and positive comments.

1. The first significant concern is that a direct assessment of endogenous CaMKII oxidation is never conducted.

It is justifiably asserted that when the single CaMKII gene in Drosophila is mutated from VV to MM, the mutant flies are now different from wild type flies because CaMKII can be oxidized in the mutants on the MM residues. The author's laboratory itself has shown (Erickson, 2008) that oxidized endogenous CaMKII can be detected in heart lysates using an antibody generated in their lab, but such immunoassays were never used in this study. There are sufficient sequence differences that might preclude cross-reactivity in flies using this antibody, so this may explain why this assay was not used in Drosophila, but it is not mentioned and should be.

In a complementary fashion, VV mice would be predicted to have diminished endogenous CaMKII oxidation in comparison to wild type mice either in skeletal muscle or in cardiac tissue as previously demonstrated (Erickson, 2008). This could be conducted in comparison to WT mice after exercise to demonstrate that this mutant is resistant to oxidation at this site. The mouse transcriptional profiles are conducted in isolated quadriceps muscle, so lysates from this tissue could also be assessed for ox-CaMKII to show more tissue specificity.

If this antibody is no longer available, another potential way in which CaMKII could be directly assessed is through CaMKII activity assays. Oxidized CaMKII should have enhanced Ca/CaM

independent activity. CaMKII activity assays are sensitive and quantitative and can be conducted on lysates of tissue in wild type and VV mice.

Another analysis that could have benefitted from anti-oxCaMKII antibody was in the fatigued mice (Fig 2H-J) including mice with the Ncf1-background. Ncf1 encodes a co-factor for NADPH oxidase, which is one enzymatic source of ROS. Thus, these mutants would be predicted to have diminished ox-CaMKII.

Thank you for these comments. Unfortunately, the anti-ox-CaMKII antiserum that we used in our original studies (e.g. Erickson, 2008)² was developed more than 12 years ago. It was distributed widely to the research community and has been depleted. Our efforts to generate more of this antiserum have so far been unsuccessful. In part, because of this, we developed and validated the CaMKII activity assay (KTR, now published in Mesubi JCI 2020¹) for use in living cells. The KTR is particularly valuable for the current studies because it allows comparison of CaMKII (VV and MM) in living cells where the relevant redox environment is intact. The KTR assay (see further comments to your points, below) indicated that MM CaMKII was more active than VV CaMKII under conditions favoring ROS generation, and that this relative increase in MM CaMKII activity was lost in the presence of antioxidant exposure (N-Acetyl-Cysteine, NAC). Based on your comments, we added new experiments that showed a similar loss of MM CaMKII activity in muscle fibers isolated from *Ncf1*^{-/-} mice (Fig. 1g-i, and Supplementary Fig. 10), lacking a functional NADPH oxidase. In order to further address your point, we assessed the solvent accessibility of wildtype and MM mutant fly CaMKII using molecular modeling, based on the human CaMKII α structure (Supplementary Fig. 3). We found that, similar to the human CaMKII, the VV residues and their replacement MM residues (located strategically in the regulatory domain) are exposed to solvent, and therefore susceptible to aqueous phase oxidation (Supplementary Fig. 3). Our modeling also showed that replacing the VV residues in fly CaMKII to MM does not alter the structure of the CaMKII regulatory or catalytic domains. Therefore, the VV to MM mutation likely preserves normal CaMKII function, consistent with our original report using CaMKII δ^2 . Finally, our in vivo data also support this model. When we minimized ROS in the climbing experiment by treating both MM and VV flies with NAC (N-Acetyl-Cysteine); The performance of MM flies regressed toward that of VV flies, while VV fly climbing rates were unaffected by NAC (Fig. 5). We interpret these findings to suggest MM- and VV-CaMKII have equivalent activity in the absence of oxidation. The observations that MM flies gain performance advantages in the presence of physiological ROS, and are impaired in the presence of excessive ROS (paraquat treatment) further support the concept that MM-CaMKII in flies indeed transduces signals through oxidation. Thus, computational, KTR, and in vivo functional data all support the concept that MM residues in fly CaMKII are oxidized and result in enhanced CaMKII activity.

The “KTR” reporter was developed and then used in mice to evaluate the ability of endogenous wild type and mutant CaMKII to influence nuclear to cytoplasmic localization changes. In supplemental figure S5, this reporter clearly translocates from the nucleus to cytoplasm in cultured cells (RPE-1) after histamine treatment and is blocked by a kinase dead CaMKII or a CaMKII inhibitor and therefore works nicely as described.

However, if it is intended to be used to assess altered CaMKII function, it is only used in a limited number of experiments (Figure 1D), the mouse myotube examples are not convincing and the n values are small, with two data points (of the 11 total) responsible for the majority of the difference. Does KTR translocate in flies?

We thank the reviewer for the positive comment on the KTR reporter developed during this study. This reporter has been further validated in our recent study in neonatal cardiomyocytes where it responded to additional ROS-inducing upstream stimuli, genetic loss of NADPH oxidase (in *Ncf1*^{-/-} mice), and a highly specific, drug-like CaMKII inhibitor (Mesubi JCI 2021, Fig. 4)¹.

In the original figure 1E (revised Fig 1g), there are indeed two data points from the MM fibers that show a particularly large response. However, if we removed these two points from the analysis, the P-value would actually become smaller (reduced from 0.0013 to p<0.0001) due to the reduction in the variance of the data, suggesting the difference before and after stimulation was not driven by these two data points. However, in order to better address your comments, and to increase the sample size, we added more data points with fibers isolated from additional mice (see revised Fig. 1g). We also replaced the original confocal images with more representative ones (Fig. 1f and Supplementary Fig. 10), and added results with *Ncf1*^{-/-} fibers. In addition, we added a new graph (Fig. 1i) to directly compare the magnitude of changes in KTR signal among different conditions.

These new experiments strengthen the data obtained in the mouse myofibers and support our conclusion that ox-CaMKII activation is critical for mobilizing intracellular Ca²⁺ in field stimulated mouse myotubes.

2. The second omission is that lifespans are not reported. It would be predicted that mutating mice and flies would alter their lifespans. In fact, the mouse knockout mutation of the MsrA gene, which encodes a reductase that can reverse the oxidation of CaMKII residues, results in extended lifespans (Ruan 2002). It is therefore predicted that VV mice would have extended lifespans and MM flies would have reduced lifespans, but neither of these were assessed for lifespan differences. Ruan H, et al (2002) PNAS (USA) 99: 2748.

We thank the reviewer for suggesting this important experiment. We performed the requested lifespan experiments with flies, taking advantage of their relatively brief lifespan and that they have only a single CaMKII gene. We tested male and female flies at 25 °C and 29 °C and found that male and female MM flies have shorter lifespan under both temperature conditions (Fig. 1d, e and Supplementary Fig. 6). Our new results thus revealed a striking effect for ROS-

sensitive CaMKII to promote aging, and strengthen support for the concept that the MM module in CaMKII provides a new example and molecular mechanism for how a gene, satisfying the precepts of antagonistic pleiotropy, can contribute to aging.

We did not perform aging studies in mice for a variety of reasons. In contrast to flies that have a single CaMKII-encoding gene, mice (and humans) have 4 CaMKII genes. Thus, it would be necessary to generate 2 new knockin lines, in addition to the CaMKII γ and δ that we have, for CaMKII α and β , perform complex and protracted husbandry followed by the longevity experiments – lasting years – to definitively rule in or rule out a role of ox-CaMKII in mouse longevity. While we agree these mouse experiments have merit, we hope you agree they are far beyond the scope of this single paper.

Other concerns:

1. In mice, gamma CaMKII is transcriptionally assessed by qPCR and shows no difference between wild type and mutant, but none of the other CaMKII genes are evaluated in mutants. This is in spite of data in the same figure (S4) showing the qPCR evaluation of all 4 mouse CaMKII genes, so primers are available. There is strong evidence in vertebrates that germ line mutation knockout of one CaMKII gene does not necessarily lead to the predicted change in CaMKII activity (Bacs (2010) PNAS 107:81; Gagnon (2014) PLOS One 9: 98186; Rothschild (2020) 742:144567).

We performed qRT-PCR to assess expression of all four CaMKII isoforms in the skeletal muscles of wildtype and VV mice, and found that the knockin replacement in *Camk2g* did not alter the relative expression of CaMKII isoforms (see revised Supplementary Fig. 8b)

2. Transcriptional responses are poorly interpreted and don't make a compelling point ("the up- or down regulation of a smaller number of genes was completely blunted or even reversed.") Fig 3D.

We improved the language (page 6-7).

3. Can ROS be measured in paraquat samples? This would also be useful confirmatory information.

Thank you for raising this issue. To address this point, we measured aconitase activity in flies treated by sucrose solution or 4 mM paraquat. Upon exposure to ROS, aconitase loses its activity, a property which has been used to assess ROS levels in flies⁴ (e.g. Das, Biochem J (2001) 360, 209-216.). We found that paraquat treatment significantly reduced the aconitase activity of both MM and VV flies, but there was no difference between genotypes either before or after treatment (revised Supplementary Fig. 5). This result suggests that there is no difference in baseline or paraquat-induced ROS between MM and VV flies. It supports the concept that MM-CaMKII increases the sensitivity of flies to ROS, without necessarily increasing ROS per se.

Reviewer #3 (Remarks to the Author):

In Wang et al, the authors look at potential antagonistic pleiotropic effects of the CAMKII protein. They find a specific two amino acid motif is fixed in vertebrates while a separate two amino acid motif is almost completely fixed in invertebrates. The vertebrate version of the protein improves skeletal muscle function in response to exercise. I found the paper interesting, and the authors appear to have done a fairly thorough job in their experimental design and execution on the effects of the CAMKII amino acid substitutions. I think this paper has the potential to be of interest to both molecular and evolutionary biologists, and I found the methods overall strong though I think a little more detail is needed to make them completely reproducible. I have listed several issues and general questions below. I also understand that with COVID, completing some of the follow-up experiments might not be possible.

Thank you for your review and positive comments.

-With regards to the interpretation that this is an “antagonistic pleiotropy” protein, I think the evidence is lacking. While improved muscle performance in flies and mice with the MM motif are shown, the negative effects of the gene are not thoroughly investigated. While the authors show the MM flies are less resistant to oxidative stress, this is not necessarily a good measure of overall fitness, especially as the oxidative theory of aging is not really supported any longer. While paraquat is often used to induce ROS in flies, this is a super, toxic stressor and is not necessarily similar to the natural increase in ROS with age seen in flies. In addition, if this increase in sensitivity to ROS is the reason invertebrates have fixed the VV sequence, why do vertebrates not also show this sensitivity and thus negative fitness that you are arguing? It would have been helpful to show a) the effects of oxidative stress on your mouse models as well as b) the effects of the MM vs VV flies on reproduction. Do the MM flies reproduce less? Have slower development? Or some other more commonly associated fitness measurement. Similarly, do the MM mice show any difference in reproduction compared to VV mice? While the authors do show that reducing ROS in the muscles of the mice reduces their performance, I think it would be better if they showed what increased ROS did the mice, as that was experiment done in the flies. If the authors can show these, I think they will have a much stronger argument for antagonistic pleiotropy at this locus.

Thank you for making these points. The case for oxidative damage in mouse disease models by virtue of the MM module in CaMKII δ has been documented extensively for cardiovascular and pulmonary diseases and diabetes by our laboratory and by others (e.g. Purohit, Circulation (2013) 128, 1749-1757; Luo, JCI (2013) 123, 1262-1274; Sanders, Science Translational Medicine (2013) 5, 195; Qu, JCI Insight (2017) 2, e90139; Swaminathan, JCI (2011) 121, 3277-3288; Wu, Sci Rep (2019) 9, 9291; reference 12, 13, 15-18 of our manuscript)⁵⁻¹⁰. Ox-CaMKII contributes to the Warberg effect, which is important for metabolic adaptations in many cancers (Hart, Nat

Commun (2015) 6, 6053; reference 14 of our manuscript)¹¹. Thus, the role of MM oxidation for triggering pathological CaMKII activation in common diseases where excessive ROS is thought to contribute to the cause or progression of disease is now well established in mice. Based on your comments, we have endeavored to make these points more clear in the revised manuscript (page 2).

The question of whether VV mice have altered reproductive performance is intriguing because CaMKII γ is essential for egg activation after fertilization in mice¹². We found that VV γ /VV γ mice are born in Mendelian ratios in heterozygous crosses, and homozygous VV γ /VV γ x VV γ /VV γ crosses produce pups normally. Our observation suggests that lacking oxidative activation of CaMKII γ does not negatively impact reproduction under laboratory conditions. On the other hand, due to their reduction in physical performance, VV γ /VV γ mice are potentially disadvantaged compared to wildtype mice in a more competitive environment, which would be interesting to determine in future studies.

Mice have 4 CaMKII genes, encoded on 4 separate chromosomes, mirroring the situation in humans. Because of this complexity, it would be very time consuming and costly to create a global VV knockin mouse (for α , β , γ and δ isoforms) that would be suitable for performing longevity studies. In contrast, flies have only a single CaMKII gene, making this model more tractable for lifespan studies. Based on your comments and the comments of other reviewers, we performed lifespan experiments with male and female flies at 25 °C and 29 °C and found that male and female MM flies have shorter lifespan under both temperature conditions (Fig. 1d, e and Supplementary Fig. 6). Thus, our new results revealed a striking effect for ROS-sensitive CaMKII to shorten lifespan. We also carried out lifetime fecundity tests, as requested. We did not find a significant difference in total progeny produced by individual MM and VV females throughout their lifespan (Supplementary Fig. 4), suggesting oxidative sensitivity of CaMKII does not impair fecundity under our laboratory conditions.

Furthermore, when carrying out crosses at 25 °C, we consistently observed that both MM and VV flies eclose on day 10, suggesting that there is no significant difference in the timing of development. We measured the size of the female flies at 5 days of age; this is the same age as the flies we used to test climbing performance. We found no difference in body size between the genotypes (Supplementary Fig. 16a), suggesting that body size difference did not contribute to climbing performance difference. We also assessed TAG stores in the same groups of flies, because muscle function can be linked to metabolic phenotype, as suggested by reviewer #1. Interestingly, we found that the MM flies have a significantly smaller TAG store (Supplementary Fig. 16b). This is interesting because CaMKII was shown to promote lipolysis, which in turn may affect exercise performance. We revised the manuscript and added relevant references related to this new finding (reference 55, 56, page 8). Taken together, these results suggest that the MM motif harms some aspects of fitness, such as resistance to ROS and longevity, but does not impair some others when evaluated under laboratory conditions. We speculate that under

natural selection in the wild, the negative impact of the MM motif interacting with the natural environment is sufficiently large to preclude its fixation in most invertebrates.

While we agree that paraquat can be a super toxic stressor, we exposed flies to a wide range of paraquat concentrations. At higher (i.e. super toxic) concentrations, sufficient to result in rapid killing, MM flies showed significantly increased mortality compared to VV counterparts. However, low concentration exposures, insufficient to reduce spontaneous movement in VV flies over several days, significantly impaired spontaneous ambulation and reduced the speed of negative geotaxic climbing in MM flies (Fig 6). Taken together with the new longevity studies in unperturbed flies, the paraquat data support a view that ROS induced debilitation and mortality are transduced by the MM module, consistent with an evolutionary trade-off.

-It appears you only used male mice while the majority of your fly experiments were done in females. Is there a reason you did not use both sexes in your study, as stronger antagonistic pleiotropic effects could be sex specific?

We used only male mice in our study because most of the studies supporting a role of the MM module in contributing to disease severity were performed in male mice, and because there is a large difference in physiological performance between males and females. Based on these considerations, we decided that it would be appropriate to focus on male mice to evaluate the role of MM-CaMKII in physiological performance. For flies, we used only females for the climbing performance and spontaneous activity. Females are better than males for these tests because males exhibit substantial sleep amounts during the day (they have a large daytime “siesta”) as well as during the night¹³. On the other hand, we performed the paraquat-induced mortality test with both males and females. Importantly, as part of this revision, we assayed the lifespan of both male and female flies and found that MM-CaMKII shortens the lifespan of both sexes. We discussed the rationale of using male mice exclusively (page 10), and using female flies for activity assays (page 11) in our revised Methods section.

-I would like to see more detail on the description of animal husbandry for both the flies and mice. Temperature? Humidity? Diet? Were the animals fed ad libitum?

We added more details in the manuscript (page 10-11).

-Is there a reason you only measured mortality at 24 and 48 hours in the paraquat flies? It would be standard to determine number dead at 8- or 12-hour intervals and calculate actual survival curves.

The paraquat assay in our study aimed to determine if MM-CaMKII confers increased sensitivity to ROS. Because we did not attempt to provide further kinetic data on mortality, we tallied the mortality only at 24 and 48 hours to keep the experiments simple. Furthermore, there is a lack

of standardization for paraquat-induced mortality in the literature. For example, Phillips et al.¹⁴ evaluated mortality at 17, 25, 38 and 48 hours, while Krůček et al. focused on 24, 48 and 72 hours¹⁵.

-You state they are in vials of 10 but how many replicate vials were used? I assume 10 replicates based on the sample sizes in the figure legends, but I would state clearly in the methods

We clarified the statement regarding sample size for negative geotaxis experiments in the revised manuscript (page 13). Specifically, "To test the negative geotaxis (climbing), females were collected and aged, as above, and kept in vials in groups of 10 flies; we tested 6-10 groups per condition in the geotaxis assay, resulting in sample sizes between 60 and 100. For paraquat or NAC treatment prior to negative geotaxis test, the flies were treated by 5% sucrose, 5% sucrose + 4 mM paraquat, 5% sucrose + 10 mM NAC, or 5% sucrose + 50 mM NAC for 24 hours at 25 °C. They were then transferred into vertical test tracks made from 25 mL serological pipette tubes using a funnel without anesthetization. During the climbing test, the flies were dislodged to the bottom of the tubes by rapidly tapping the vials on the desktop for 10 times and climbing was video recorded for subsequent analysis. Each group of flies was tested for 10 consecutive trials at 30 seconds intervals. The vertical distances the flies climbed in 6 seconds since the last tap (time 0) were used to calculate the vertical velocity of climbing. Flies that initiated flight or paused during the 6-second time window were excluded from the analysis. Because of this exclusion criterion, the final sample sizes were different from multiples of 10, and the specific n's were presented in the figure legends."

-Please provide a statistical analysis section in the methods. In some of the figure legends you mention the stats used, but in others you do not. It would be much clearer if you would just state all the analyses used in the methods. In addition, did you make sure that your data met the assumptions of the different statistical tests used?

We added a statistical analysis section in the revised Methods section (page 20). Based on your comments, we reanalyzed our data by first carrying out normality tests (D'Agostino-Pearson) when applicable. When the data did not pass the normality test, we performed nonparametric tests. In addition, we now carried out all parametric tests without assuming equal variance. We updated all *P*-values in figures to reflect these changes. None of the changes affect the conclusions made in the manuscript.

-Looking at Table S1, it appears that the only vertebrates without the MM/CM module are within the ray finned fishes. Any thoughts as to why this group might have lost this specific module? In addition, are the 27 fish species without the module all sister species, such that the loss occurred once on the phylogeny?

Ray-finned fish have eight or more copies of CaMKII genes, likely due to genome/gene duplication. No individual species of the ray-finned fish have entirely lost the CM/MM module

in all CaMKII isoforms; typically, only one CaMKII isoform has lost the module in a single species. The increased redundancy of CaMKII genes in these species might have permitted CM/MM module loss. However, we cannot be sure that all CaMKII isoforms in ray-finned fish are expressed and functional without further, extensive studies. The alternative modules in ray-finned fish include VV, IL, MV, MI, KM, and SM. The diversity of these modules, as well as the topology of the teleost tree, suggest the loss happened multiple times independently. Determining exactly how many losses occurred and under what conditions will require much more sampling within this incredibly diverse radiation. It will be interesting to study the evolutionary implication of these events in future studies – it likely represents an exciting research program in-and-of itself. We added this comment on the ray-finned fishes to the footnote of Supplementary Table 1.

-On page 4 line 6, I believe you have the WT next to the wrong genotype.

Thank you for your careful reading. We corrected this error.

-Along the same lines, I found the constant reference to WT to be confusing. I think you clearly laid out that MM is the WT in mice while VV is the WT in flies, and I would remove the constant (WT) reference.

Thank you for making this point. We respectfully ask to keep this distinction. Although redundant, the distinction between MM/WT mice and VV/WT flies is critical to reading and interpreting our work, and we are concerned that its loss may confuse some readers.

-Please include the ages of the animals used in experiment. You state 13-15 weeks in the RNAseq analysis, was this the age for all the other mouse experiments as well?

We included the ages of all the animals in the revised Methods.

-Your mice appear to only be around 10% body fat (Figure S6), this seems really low compared to most B6 mouse studies, as well as most other strains. Is this due to young age or potentially a husbandry effect?

Thanks for highlighting this observation. We checked the Jackson laboratory's phenotyping web pages, and found that mice characterized in our facility (compared with data from the Jackson laboratory) are on the low side of the normal range for body weight for their age, and are indeed lower in fat percentage (The Jackson laboratory's male mice were 15.5% fat at 8 weeks of age, vs. ours at 10.26%). These differences could be due to husbandry conditions, as you suggested. However, we cannot exclude the possibility that there are differences in echo-MRI machines. We noted that in Jackson Laboratory's data, the mass unaccounted for, which should be bones, is only about 8% of body weight. This number is typically higher when measured in

our facility (>10%). As examples, we compared our data to the measurements made by our core from two recent cohorts of C57BL/6J mice. Our data are comparable to these cohorts: A cohort from February, 2020 (purchased from the Jackson Laboratory). Same age as our mice (11-12 weeks), same male sex, same Teklad diet: (n=8) body weight = 24.6 +/- 0.8 g , fat = 2.73 +/- 0.33 g , 11.07% fat, 77.17% lean (avg +/- se), which are similar to ours mice. Ours: 25.8 +/- 0.47 g, fat = 2.63 +/- 0.11 g, 10.26% fat, 76.9% lean. Another cohort also characterized in February 2020: WT on a B6 background, (n=12) 25.3 +/- 0.68 g, fat = 2.57 +/- 0.1 g, 10.18 % fat, 78.08% lean, also similar to ours.

-Page 4 line 11, you state the reduced exercise performance in VV mice could be due to “metabolism or motivation”. Did you measure metabolism or behavior in these animals? While measuring motivation is difficult, if not impossible, other basic behavioral measures, besides voluntary wheel running, might provide some insights into the differences with these two genotypes.

We measured glucose and insulin tolerance of the mice as shown in revised Supplementary Fig. 12. Additionally, we assessed oxygen consumption rate, respiratory control ratio and energy expenditure in CLAMS in the presence of running wheels, and we found no difference between the MM and VV mice; these data are now also included in Supplementary Fig. 12. For behavior measures, we measured voluntary wheel running as an index of motivation. However, due to COVID-19, we did not perform additional behavioral experiments.

*-The *ncf1* mice are not listed in your methods at all. Please include where these came from, ages, sex studied, etc.*

We added the description about *Ncf1*^{-/-} mice to the revised Methods section (page 12).

-Page 2 line 10 and page 3 line 25 use the word “enjoy”. However, this word is not correct for evolutionary derived characters, please change.

Thank you for making this point. We corrected the wording.

-I found it interesting that you found differences in inflammation between the MM and VV mice. As low grade inflammation is now thought to be a potentially large contributor to aging, have you aged out these mice to see if the VV mice age differently than the MM mice? If MM tends to promote inflammation, potentially the VV mice are longer lived? This is most likely its own study, but I think it might be nice for the authors to hypothesize how this inflammatory response might effect aging in the discussion. Along the same lines, are their differences in longevity in the VV and MM flies? This would be a much simpler/quicker experiment.

This is an interesting point that we have considered. As noted above, mice have 4 CaMKII genes, encoded on 4 separate chromosomes, mirroring the situation in humans. Because of this complexity, it would be very time consuming and costly to create a global VV knockin mouse (for α , β , γ and δ isoforms) that would be suitable for performing longevity studies for this revision. We will consider this study in the future. Also, as discussed above, we found unperturbed MM flies had shorter lifespans than VV flies, as you suggest. We added a discussion of the potential roles of oxidized CaMKII to drive aging through promoting inflammation (page 10).

Reviewer #4 (Remarks to the Author):

Wang et al study the interesting case of CaMKII, a signaling molecule with a pleiotropic effects. Specifically, they analyze how the vertebrate specific acquisition of an (Reactive) oxygen-sensing domain led to outstanding benefits in muscle performance, but led to the increased risk of cardiac failure in late life. Hence, CaMKII would be represent an exquisite candidate to fit the predictions of the antagonistic pleiotropism theory of aging, proposed by George Williams. The manuscript is well written and presents a large amount of compellind results.

Thank you for your review and positive comments.

However, there are some aspects that need careful consideration. The main assumption of the manuscript is that the radical theory of aging is correct and ROS are bad. However, the impact of ROS on aging has been completely reconsidered over the past years in light of the theories of hormesis (e.g. mitormesis). ROS would trigger "rescue" modes in organisms, which would benefit healthspan. This should be discussed and the results presented should be framed also in light of hormesis.

Thank you for making this point. We agree that ROS are likely to play nuanced roles in aging, in part due to the competing beneficial hormetic responses to ROS. This is why we proposed that “The fact that ROS contribute to aging and aging-related diseases, but are also capable of beneficial roles suggests that proteins responsible for ROS-sensing might be part of an antagonistic pleiotropic genetic program that drives aging-related diseases, including cardiovascular disease and cancer” (page 2).

Thus, we agree that our study could also be framed in light of hormesis because the physiological level of ROS confer performance benefits in MM flies, while pathological ROS leads to frailty and death. Furthermore, MM flies show elevated expression of antioxidant genes such as GstD2 and Sulfiredoxin (Fig. 4C), consistent with the concept of ROS-related hormesis. In order to make these points more clearly, we have added a brief discussion of hormesis in the revised manuscript (page 10) and included a new diagram (Fig. 7).

Conceptually, in the introduction there is a slight misconception in that all the genetic variants present in nature are there because of the action of natural selection. However, there is a lot of genetic variation that has nothing to do with strong purifying selection (lines 28-29 of page 2).

Thank you for raising this issue. We certainly did not intend to promote a pan-selectionist paradigm or convey that theoretical stance. We expanded the text to discuss the role that stochastic processes play in antagonist pleiotropy theory and to demonstrate that drift was considered as a potentially important driver of these phylogenetic patterns (page 2).

Also, the authors did not mention the mutation antagonistic (MA) theory of aging. We encourage them to cite it and to speculate whether their observations would fit MA, rather than AP.

Thank you. We did initially consider this point. However, the mutation accumulation theory of Medawar is a model of aging in which the age-dependent reduction of selective pressure permits the accumulation of deleterious mutations that are expressed late in life. This model is incongruent with our data in that the randomized evolution it predicts for the involved loci should also produce a highly randomized sequence at those loci when sampled across a taxonomically inclusive tree – a sampling prediction that would include losses of deleterious traits due to drift as well as their accumulation. This prediction is in sharp contrast to the highly conserved sequence that we find and that suggests the locus is being actively maintained by selection. Since that selection is unlikely to be of the negative variety because of the selection shadow that is a component of both the MA and AP models, then the positive selection that is a tenet of AP but not MA is the more reasonable explanation. Regardless, this was an important point to cite and discuss, which we did in the revised manuscript (page 2).

I would find very interesting to see multiple alignments for CaMKII across different mammals. This should be pretty straightforward to do. The authors could show how conserved this domain is in species with different degrees of muscle performance and thermal requirements. How conserved is the MM/MC domain across mammals? The strong amino acid conservation of the CM/MM module is indicative for a strong purifying selection and could be preceded by initial positive selection in early vertebrates leading to fixation. Although the positive selection signal might be obscured by the rapid fixation and consequent purifying selection since stem vertebrates, measuring the strength of selection using Dn/Ds for the CaMKII genes would strengthen the argument.

Thank you for raising these points. We have now included all the alignments of CaMKII in the source data files, including a number of different mammalian taxa representing a variety of mammalian lineages and behavioral ecologies. The alignments show that CM/MM module is completely conserved among all the mammalian species we examined (488 sequences from placentals, 16 from marsupials, and 4 from a monotreme), regardless of muscle performance or thermal requirement. The sampling strategy is, of course, far from exhaustive but when considered in light of the similarly conserved nature of this locus in all the other vertebrate lineages (besides teleost fish) there is little reason to expect that more extensive sampling

within mammals would produce significant variation capable of altering the major evolutionary trends we describe. We speculate that the CM/MM module likely plays additional roles, beyond regulating motor function or immune responses because CaMKII is a multifunctional enzyme expressed throughout the body. Any anomalous variation (in mammals or other clades) would be highly interesting and helpful in studying these more specific hypotheses; but this is well outside the scope of this single paper.

We appreciate the reasonable suggestion to calculate a Dn/Ds ratio. The problem is that our sampling failed to reveal any non-synonymous substitutions for the regulatory loci responsible for the oxidative pathway in all of vertebrates – the only exception being within teleost fish, which is far removed from the phylogenetic backbone connecting mice, humans, flies (and that encompasses the vast majority of bilaterian diversity). So there really isn't a non-zero value ratio to calculate. It would be interesting to know just how many synonymous substitutions are present at these loci, but unless that number is also zero (which is extremely unlikely) then positive/purifying selection is going to be strongly supported. If that number did somehow turn out to be zero (which would probably then require an even more extensive sampling to convince people of this), the conclusion wouldn't be that the sequence is evolving randomly (as under a mutation addition model) but rather that some more inclusive constraint is in place – inclusive in the sense of involving a longer section of the genome. If this were the case, then positive selection would still be inferred but it would not necessarily be targeting the oxidative pathway. Here again, additional analyses and experimentation would likely be necessary to support any robust conclusions; certainly well worth doing but seemingly outside the scope of this particular paper.

The transcription response upon 10mM paraquat shown in Figure 4 is significantly higher in MM vs. VV genotypes, but overall the direction of response is very similar between MM and VV. The ROS-signaling MM genotype seems to induce a stronger response of the same transcriptional machineries. Could the dose dependent response be responsible for the detrimental effects of the MM genotype late in life?

Thank you for making this point. Yes. The hyperactivation of CaMKII through oxidation in the presence of excessive ROS is likely the reason for the detrimental effects of MM-CaMKII. This is consistent with the dose-dependent, hormesis roles of ROS. We added a diagram to further elaborate this point (Fig. 7). Furthermore, we agree that MM- and VV-CaMKII may mobilize the same transcriptional machineries to different degrees. We added a comment about this point when presenting the transcriptional data of mouse muscles (page 6 line 44 to page 7 line 11).

The most critical aspect of the whole paper is that indeed the CaMKII MM domain is selected for by natural selection (it improves fitness) but it remains harmful late in life, representing a risk for aging/death. The authors refer to extensive literature in this respect, connecting CamKII function/activation during stroke and cardiac failure. However, they do not show (or maybe I

missed it?) this in their own experimental work in mice. I could not see evidence that VV is protective over MM in terms of (for instance) stroke risk or cardiac failure. In the parts where it should demonstrate AP, the study becomes needlessly complicated, yet fails to provide sufficient evidence. It has shown that MM mice outperform VV mice; but do VV outperform MM mice in some viability metric (or demonstrate a deceleration of the aging phenotype) under usual circumstances. If VV is protective to aging-related damage in mice, based on the authors own work, I would be very impressed and be convinced by their argument.

Thank you for making these points. The case for oxidative damage in mouse disease models by virtue of the MM module in CaMKII δ (the predominant isoform in heart and some other tissues) has been documented extensively for cardiovascular and pulmonary diseases and diabetes by our laboratory and by others⁵⁻¹⁰ (cited as references 12, 13, 15-18 of our manuscript). Furthermore, ox-CaMKII contributes to the Warberg effect, which is important for metabolic adaptations in many cancers¹¹ (reference 14 of our manuscript). Thus, the role of MM oxidation for triggering pathological CaMKII activation that contributes to common diseases where excessive ROS is thought to contribute to the cause or progression of disease in mice is now well established. Based on your comments, and the comments of reviewer 2, we have endeavored to make these points more clear in the revised manuscript (page 2).

Fly work: The adverse effects of the CM/MM module shown in flies are very strong. My concern is that the levels of paraquat used in the experiments are not comparable to normal physiological levels. Can the author comment on this and if necessary, clarify this in the manuscript. The MM genotype in flies seems to perform better in climbing, shortening velocity and relaxation rate in flies (Figure 6). This effect is diminished when adding paraquat, a ROS inducing toxin. The fact that there is an advantage for the MM genotype in flies, but it was not fixed in the invertebrate lineage would implicate that invertebrates maybe encountered higher level of ROS in their environment? How is the performance of climbing in mice when exposed to higher ROS levels? Is the concluded trade-off connected to a benefit in a feature that is missing in invertebrates like skeletal muscle or maybe even adaptive immunity?

Based on your comments and suggestions from other reviewers, we performed longevity studies with flies at both 25 °C and 29 °C, without artificially manipulating ROS. We found that MM flies have a shorter lifespan under both temperature conditions (revised Fig 1d, e and Supplementary Fig. 6). Thus, our new results revealed a striking effect for ROS-sensitive CaMKII to shorten lifespan in the presence of endogenous ROS. It is indeed possible that MM was not fixed in the invertebrate lineage because this lineage encounters more ROS, or have lower antioxidant capacity. Furthermore, it is also possible that the fixation of the CM/MM module requires additional adaptations that are vertebrate-specific, such as adaptive immunity, as you suggested. We added this point to the Discussion in the manuscript (Discussion, last paragraph). While we hope to study these points in the future, we submit they are beyond the scope of the present work.

We initially considered performing a treadmill test with mice after treating them with paraquat. However, we did not carry out this study because there are four CaMKII genes in mice, and all are ROS-sensitive, vastly complicating the interpretation of such a study. Furthermore, it would be prohibitive to generate the 4 MMVV knockin mice and subsequently perform the husbandry to create a 'global' VV mouse (i.e. with knockins to α , β , γ and δ CaMKII isoforms).

There are some typos and parts that are unclear:

- page 4, line 6: MM and WT(VV). It should be MM(WT) and VV.

Thank you. These were corrected (page 5)

- page 4, line 13: fig. 7Aa does not exist. Is it perhaps fig. S7A?

Thank you. These were corrected.

- Figure 3: what is the difference between E and F? I could not grasp

Sorry for this confusion. We have improved our description in the caption.

- Figure 3 should also show what's up-regulated in VV vs. MM.

We added a panel (revised Fig. 3e) to show the expression pattern of genes whose up- or down-regulation reached significance ($q < 0.05$) only in the VV muscles in response to exercise, similar to our original Fig. 3d. The newly added Fig. 3e further strengthened the conclusion that MM and VV muscles respond to exercise in a qualitatively similar fashion, revealing most differences to be quantitative. We speculated that CaMKII oxidation regulates global gene expression primarily by augmenting the Ca^{2+} signals in muscles during exercise (as shown in Fig. 1h-j). Because of the conserved CM/MM motif in all CaMKII isoforms we speculate that ox-CaMKII may augment gene transcription in diverse tissues. We added this discussion into our revised manuscript (page 6 line 44 to page 7 line 11).

** See Nature Research's author and referees' website at www.nature.com/authors for information about policies, services and author benefits.

- 1 Mesubi, O. O. *et al.* Oxidized CaMKII and O-GlcNAcylation cause increased atrial fibrillation in diabetic mice by distinct mechanisms. *Journal of Clinical Investigation* **131**, doi:10.1172/jci95747 (2021).
- 2 Erickson, J. R. *et al.* A dynamic pathway for calcium-independent activation of CaMKII by methionine oxidation. *Cell* **133**, 462-474, doi:10.1016/j.cell.2008.02.048 (2008).
- 3 Konstantinidis, K. *et al.* MICAL1 constrains cardiac stress responses and protects against disease by oxidizing CaMKII. *Journal of Clinical Investigation* **130**, 4663-4678, doi:10.1172/jci133181 (2020).
- 4 Das, N., Levine, R. L., Orr, W. C. & Sohal, R. S. Selectivity of protein oxidative damage during aging in *Drosophila melanogaster*. *Biochem J* **360**, 209-216, doi:10.1042/0264-6021:3600209 (2001).
- 5 Purohit, A. *et al.* Oxidized Ca(2+)/calmodulin-dependent protein kinase II triggers atrial fibrillation. *Circulation* **128**, 1748-1757, doi:10.1161/CIRCULATIONAHA.113.003313 (2013).
- 6 Luo, M. *et al.* Diabetes increases mortality after myocardial infarction by oxidizing CaMKII. *J Clin Invest* **123**, 1262-1274, doi:10.1172/JCI65268 (2013).
- 7 Sanders, P. N. *et al.* CaMKII is essential for the proasthmatic effects of oxidation. *Science translational medicine* **5**, 195ra197, doi:10.1126/scitranslmed.3006135 C2 - PMC4331168 (2013).
- 8 Qu, J. *et al.* Oxidized CaMKII promotes asthma through the activation of mast cells. *JCI Insight* **2**, e90139, doi:10.1172/jci.insight.90139 (2017).
- 9 Swaminathan, P. D. *et al.* Oxidized CaMKII causes cardiac sinus node dysfunction in mice. *J Clin Invest* **121**, 3277-3288, doi:10.1172/JCI57833 (2011).
- 10 Wu, Y., Wang, Q., Feng, N., Granger, J. M. & Anderson, M. E. Myocardial death and dysfunction after ischemia-reperfusion injury require CaMKII δ oxidation. *Sci Rep* **9**, 9291, doi:10.1038/s41598-019-45743-6 (2019).
- 11 Hart, P. C. *et al.* MnSOD upregulation sustains the Warburg effect via mitochondrial ROS and AMPK-dependent signalling in cancer. *Nat Commun* **6**, 6053, doi:10.1038/ncomms7053 (2015).
- 12 Backs, J. *et al.* The gamma isoform of CaM kinase II controls mouse egg activation by regulating cell cycle resumption. *Proc Natl Acad Sci U S A* **107**, 81-86, doi:10.1073/pnas.0912658106 (2010).
- 13 Huber, R. *et al.* Sleep homeostasis in *Drosophila melanogaster*. *Sleep* **27**, 628-639, doi:10.1093/sleep/27.4.628 (2004).
- 14 Phillips, J. P., Campbell, S. D., Michaud, D., Charbonneau, M. & Hilliker, A. J. Null mutation of copper/zinc superoxide dismutase in *Drosophila* confers hypersensitivity to paraquat and reduced longevity. *Proceedings of the National Academy of Sciences* **86**, 2761-2765, doi:10.1073/pnas.86.8.2761 (1989).
- 15 Krůček, T. *et al.* Effect of low doses of herbicide paraquat on antioxidant defense in *Drosophila*. *Archives of Insect Biochemistry and Physiology* **88**, 235-248, doi:10.1002/arch.21222 (2015).

Reviewers' Comments:

Reviewer #1:

Remarks to the Author:

The authors have addressed my concerns and I recommend the publication of the manuscript in Nature Communications.

Reviewer #2:

Remarks to the Author:

Re-Review of Manuscript # NCOMMS-20-24290A

The Wang et al. manuscript entitled "A Critical Performance/Disease Trade-off at the Dawn of Vertebrate Evolution" has been resubmitted as "CaMKII Oxidation is a Critical Performance/Disease Trade-off Acquired at the Dawn of Vertebrate Evolution."

The abstract of this manuscript has not changed substantially, but the authors have comprehensively addressed all of the comments and suggestions by the reviewers, improving the text and flow of the manuscript and adding several new supportive figures. These figures include additional life-span measurements in flies and are convincing (Fig 1,d,e and Supp Fig 6). Other data sets have been improved by extra sampling making the findings more convincing. In the meantime, a paper was published which validates the CaMKII activity reporter used and referenced in this manuscript. Effects on other CaMKII genes were conducted (Supp Fig 8b), which had not been conducted before (Supp Fig 4b). These addressed a potentially confounding phenomenon of gene compensation. There were many other changes, all of which contributed to a much more compelling manuscript.

Other minor Comments:

1. Can you Mention in Methods section on the validation of the CaMKII activity sensor (CaMKII-KTR) that the antibody used to measure CaMKII Ox is no longer available.
2. In the manuscript, can you elaborate on gender-specific differences in survivability at 25C (Fig1d,e) but not at 29C (Suppl. Fig. 6)?

Reviewer #3:

Remarks to the Author:

I think the authors have greatly improved their manuscript, and they addressed the reviewers concerns well. The only thing I would have liked to see was at least a couple "limitations" in the discussion section. For example, while I understand why the authors were unable to do lifespan/aging studies in the mice (4 copies and time), it would be helpful for them to mention this in the discussion.

Reviewer #4:

Remarks to the Author:

I find the authors' responses and the revised manuscript satisfactory and I congratulate the authors for the work done.

Reviewer #5:

Remarks to the Author:

Overall, I very much enjoyed reading this study. The concept of antagonistic pleiotropy and the

example of oxidative activation of CaMKII are very elegant.

The abstract is very interesting and nicely written. I think this study will be of interest to a wide range of readers and investigators.

The authors hypothesize that "gaining the MM module allowed ROS to enhance skeletal muscle performance through ox-CaMKII"

My comments focus on the analysis of exercise performance and muscle function in the mouse models.

I think the exercise tests are sound and the MM mice do exhibit an increase in running capacity.

The data showing a change in stimulated calcium transients is also convincing and points toward this CaMKII-dependent calcium mechanism regulating fatigue of the fibers. Was the decay in force (fatigue resistance) also measured in these isolated fibers. This would definitely demonstrate a muscle intrinsic effect.

Several pieces of additional evidence are presented to determine if this change in CaMKII activity is muscle intrinsic or the fatigue resistance is due to extrinsic factors

Muscle fatigue measured in vivo by torque analysis of the quadriceps suggests that it is intrinsic to the muscle but, does not rule out the contribution of metabolic factors (nutrients and oxygen) delivered by the circulation or changes in neuronal function.

Additional experiments were performed with single fibers isolated from the FDB. In these fibers the calcium handling was evaluated and the presented change in calcium transients supports a muscle intrinsic response.

The investigators also used a CaMKII reporter that translocated from the nucleus to the cytoplasm upon electrically stimulated contraction of the fiber. The nuclei lose their fluorescent intensity which I presume indicates the signal has translocated to the myofiber cytoplasm.

Several additional whole-body measures were reported in the supplement. These include muscle and body weights; blood lactate levels glucose tolerance and all of these variables were not different between the two strains.

There were trends in the whole-body oxygen consumption rate (OCR), $p=0.0529$, RER $p=0.0741$ and EE, $p=0.0509$ that were measured in $n=11$ MM mice and $n=8$ VV mice. These are not statically different as the authors report. Is the sample size large enough for this experiment? Skeletal muscle makes up a large percentage of overall body composition the rate of whole body oxygen consumption could be altered.

Fiber type composition is also analyzed in muscle cross section using myosin heavily chain antibodies. It is noted that the percent of each fiber type was counted only a portion of the muscle cross-sectional area. Fiber types counts should be systematically, randomly counted across the total muscle area or 100% of the fibers should be counted in the cross section.

Mitochondrial respiration was also measured in isolated fibers that were cultured on laminin coated plates using a Seahorse system. How were these values normalized? The graph indicates the number of fibers – are these of different lengths with different amounts of mitochondria. These types of assays are usually performed in permeabilized fiber bundles and respiration can be monitored over more physiologic ranges in Complex I- IV. Did any of the fibers hyper-contract or were damaged

(cytochrome C response) in this system? This could change the rate of respiration. This experiment does not add much to the overall study.

Reviewer #1 (Remarks to the Author):

The authors have addressed my concerns and I recommend the publication of the manuscript in Nature Communications.

We thank the viewer for recommending the publication of the revised manuscript, and the constructive suggestions that helped us to greatly improve the original manuscript.

Reviewer #2 (Remarks to the Author):

Re-Review of Manuscript # NCOMMS-20-24290A

The Wang et al. manuscript entitled “A Critical Performance/Disease Trade-off at the Dawn of Vertebrate Evolution” has been resubmitted as “CaMKII Oxidation is a Critical Performance/Disease Trade-off Acquired at the Dawn of Vertebrate Evolution.”

The abstract of this manuscript has not changed substantially, but the authors have comprehensively addressed all of the comments and suggestions by the reviewers, improving the text and flow of the manuscript and adding several new supportive figures. These figures include additional life-span measurements in flies and are convincing (Fig 1,d,e and Supp Fig 6). Other data sets have been improved by extra sampling making the findings more convincing. In the meantime, a paper was published which validates the CaMKII activity reporter used and referenced in this manuscript. Effects on other CaMKII genes were conducted (Supp Fig 8b), which had not been conducted before (Supp Fig 4b). These addressed a potentially confounding phenomenon of gene compensation. There were many other changes, all of which contributed to a much more compelling manuscript.

We thank the reviewer for the positive comments, and the constructive suggestions that helped us greatly improve the original manuscript.

Other minor Comments:

1. Can you Mention in Methods section on the validation of the CaMKII activity sensor (CaMKII-KTR) that the antibody used to measure CaMKII Ox is no longer available.

We added this information as requested (page 15).

2. In the manuscript, can you elaborate on gender-specific differences in survivability at 25C (Fig1d,e) but not at 29C (Suppl. Fig. 6)?

We noted that male flies are more susceptible than females to the presence of the MM module at both 25 °C and 29 °C. Furthermore, under the condition of heat stress at 29 °C, the detrimental effects of the

MM module further diminish in females. This sex difference may reflect the difference in ROS scavenging capacities between sexes. Specifically, females have a better ROS scavenging capacity during aging¹. Thus, females may have less ROS to hyperactivate MM-CaMKII during aging and thus tolerate the MM module better. We speculate that because MM-CaMKII contributes less to aging in females, its detrimental effect is largely masked in females at 29 °C, where heat stress may accelerate aging through mechanisms in addition to oxidative stress.

We added this discussion and the reference to the manuscript (page 4)

Reviewer #3 (Remarks to the Author):

I think the authors have greatly improved their manuscript, and they addressed the reviewers concerns well. The only thing I would have liked to see was at least a couple "limitations" in the discussion section. For example, while I understand why the authors were unable to do lifespan/aging studies in the mice (4 copies and time), it would be helpful for them to mention this in the discussion.

We thank the viewer for the positive comments, and the constructive suggestions that helped us to greatly improve the original manuscript.

We added the limitations about not having aging tested on mice in the discussion (page 10).

Reviewer #4 (Remarks to the Author):

I find the authors' responses and the revised manuscript satisfactory and I congratulate the authors for the work done.

We appreciate your positive comments and thank you for your constructive suggestions on our original manuscript.

Reviewer #5 (Remarks to the Author):

Overall, I very much enjoyed reading this study. The concept of antagonistic pleiotropy and the example of oxidative activation of CaMKII are very elegant.

The abstract is very interesting and nicely written. I think this study will be of interest to a wide range of readers and investigators.

We thank you for these positive comments.

The authors hypothesize that “gaining the MM module allowed ROS to enhance skeletal muscle performance through ox-CaMKII”

My comments focus on the analysis of exercise performance and muscle function in the mouse models.

I think the exercise tests are sound and the MM mice do exhibit an increase in running capacity.

We thank you for the positive comments.

The data showing a change in stimulated calcium transients is also convincing and points toward this CamKII-dependent calcium mechanism regulating fatigue of the fibers. Was the decay in force (fatigue resistance) also measured in these isolated fibers. This would definitely demonstrate a muscle intrinsic effect.

We agree that directly measuring the decay in force in isolated fibers would further validate the role of ox-CaMKII conferring fatigue resistance. Unfortunately, we are not equipped to measure fiber contractility directly, so this experiment was not performed. However, we recognize the correlation between calcium transient size and fiber contractility is well-established²⁻⁵, so our calcium transient data strongly suggest an inferior fatigue resistance of the VV muscle fibers. We hope the reviewer would agree that measuring fatigue resistance at the muscle fiber level is a technically heroic task that would require adapting and validating a new technology or establishing a new collaboration, and, in any event, is not necessary for the validity of this manuscript.

Several pieces of additional evidence are presented to determine if this change in CamKII activity is muscle intrinsic or the fatigue resistance is due to extrinsic factors

Muscle fatigue measured in vivo by torque analysis of the quadriceps suggests that it is intrinsic to the muscle but, does not rule out the contribution of metabolic factors (nutrients and oxygen) delivered by the circulation or changes in neuronal function.

We thank the review for pointing out this important caveat. We acknowledged the potential involvement of ox-CaMKII extrinsic to muscles in the manuscript (page 6). Furthermore, our manuscript proposed that “ox-CaMKII plays diverse physiological roles, beyond those uncovered by this study in the skeletal muscles and mast cells” (page 10, line 17). Thus, we believe that it is not imperative, nor is it possible in this foundational and already substantial paper, to rule-in or rule-out the role of ox-CaMKII outside of skeletal muscles for the conclusion to stand.

Additional experiments were performed with single fibers isolated from the FDB. In these fibers the calcium handling was evaluated and the presented change in calcium transients supports a muscle intrinsic response.

We thank the reviewer for the positive comments.

The investigators also used a CamKII reporter that translocated from the nucleus to the cytoplasm upon

electrically stimulated contraction of the fiber. The nuclei lose their fluorescent intensity which I presume indicates the signal has translocated to the myofiber cytoplasm.

You are correct. The CaMKII-KTR reports CaMKII activity by shuttling between the nuclei and the cytosol (please see Supplementary Fig. 9 for more detail). Briefly, we built a CaMKII interacting domain into the reporter to confer CaMKII-specificity. When cellular CaMKII activity is elevated, the build-in nuclear localization signal (NLS) and nuclear export signal (NES) peptides in the reporter are phosphorylated by CaMKII. The phosphorylation alters the relative strengths of the NLS and NES⁶, resulting in a net increase in nuclear export of the reporter. The cyto/nuc fluorescence is expressed as a ratio because of its mechanism of action. Use of this reporter has previously been reported in mouse neonatal cardiomyocytes (figure 4 of reference⁷) and characterized in RPE-1 cells in this manuscript. For the muscle fiber experiments, we used the cytosolic/nuclear signal ratio to index CaMKII activity as reported before^{6,7}, which minimizes artifacts due to photobleaching.

Several additional whole-body measures were reported in the supplement. These include muscle and body weights; blood lactate levels glucose tolerance and all of these variables were not different between the two strains.

There were trends in the whole-body oxygen consumption rate (OCR), $p=0.0529$, RER $p=0.0741$ and EE, $p=0.0509$ that were measured in $n=11$ MM mice and $n=8$ VV mice. These are not statically different as the authors report. Is the sample size large enough for this experiment? Skeletal muscle makes up a large percentage of overall body composition the rate of whole body oxygen consumption could be altered.

We agree with the reviewer that the whole-body oxygen consumption and respiratory control exchange ratio (RER) could differ between the MM and VV mice. It is interesting because, during the revision, we observed that the MM flies have reduced triglyceride stores compared to VV flies, suggesting that CaMKII oxidation can influence whole-body metabolism. Our sample size ($n=11$ and $n=8$) provides about 80% power to detect a difference in 1.5 SD or greater. Detecting a smaller effect size would require a larger sample size. We hope to address whether CaMKII oxidation would induce a subtle but significant difference in metabolism in future studies; we believe this future investigation is outside of the scope of the present study.

Fiber type composition is also analyzed in muscle cross section using myosin heavily chain antibodies. It is noted that the percent of each fiber type was counted only a portion of the muscle cross-sectional area. Fiber types counts should be systematically, randomly counted across the total muscle area or 100% of the fibers should be counted in the cross section.

Thank you for making this point. We counted all (100%) fibers through the entire cross section of each quadriceps muscle. We revised the legend of Supplementary fig. 15b to make this point clearer.

Mitochondrial respiration was also measured in isolated fibers that were cultured on laminin coated plates using a Seahorse system. How were these values normalized? The graph indicates the number of

fibers – are these of different lengths with different amounts of mitochondria. These types of assays are usually performed in permeabilized fiber bundles and respiration can be monitored over more physiologic ranges in Complex I- IV. Did any of the fibers hyper-contract or were damaged (cytochrome C response) in this system? This could change the rate of respiration. This experiment does not add much to the overall study.

Thank you for raising these points. We normalized the values to the number of muscle fibers counted under the microscope. We used this approach because the small number of fibers precludes accurate protein quantification. We isolated the muscle fibers under identical conditions from age-matched mice, and we did not observe noticeable differences in fiber size, nor did we see hyper-contraction during the experiments. We used intact fibers because they may better preserve the effects of ox-CaMKII, which would be lost if we permeabilize the fibers. We agree with the reviewer that this approach precludes detailed studies on the function of individual mitochondrial respiratory complexes. However, because the RNA sequencing study (Fig. 3) and Western blot (Fig. 14a) did not reveal significant differences in mitochondrial content between genotypes, we did not further pursue mitochondrial assays in this initial paper. We acknowledge that it is formally possible that ox-CaMKII might regulate oxidative phosphorylation or glycolysis under more physiological conditions.

References

- 1 Niveditha, S., Deepashree, S., Ramesh, S. R. & Shivanandappa, T. Sex differences in oxidative stress resistance in relation to longevity in *Drosophila melanogaster*. *Journal of Comparative Physiology B* **187**, 899-909, doi:10.1007/s00360-017-1061-1 (2017).
- 2 Allen, D. G., Lamb, G. D. & Westerblad, H. Skeletal muscle fatigue: cellular mechanisms. *Physiological Reviews* **88**, 287-332, doi:10.1152/physrev.00015.2007 (2008).
- 3 Melzer, W., Herrmann-Frank, A. & Lüttgau, H. C. The role of Ca²⁺ ions in excitation-contraction coupling of skeletal muscle fibres. *Biochimica et Biophysica Acta (BBA) - Reviews on Biomembranes* **1241**, 59-116, doi:10.1016/0304-4157(94)00014-5 (1995).
- 4 Berchtold, M. W., Brinkmeier, H. & Müntener, M. Calcium Ion in Skeletal Muscle: Its Crucial Role for Muscle Function, Plasticity, and Disease. *Physiological Reviews* **80**, 1215-1265, doi:10.1152/physrev.2000.80.3.1215 (2000).
- 5 Westerblad, H. & Allen, D. G. Changes of myoplasmic calcium concentration during fatigue in single mouse muscle fibers. *Journal of General Physiology* **98**, 615-635, doi:10.1085/jgp.98.3.615 (1991).
- 6 Regot, S., Hughey, J. J., Bajar, B. T., Carrasco, S. & Covert, M. W. High-sensitivity measurements of multiple kinase activities in live single cells. *Cell* **157**, 1724-1734, doi:10.1016/j.cell.2014.04.039 C2 - PMC4097317 (2014).
- 7 Mesubi, O. O. *et al.* Oxidized CaMKII and O-GlcNAcylation cause increased atrial fibrillation in diabetic mice by distinct mechanisms. *J Clin Invest* **131**, doi:10.1172/JCI95747 (2021).

Reviewers' Comments:

Reviewer #5:

Remarks to the Author:

The authors have nicely responded to all the comments. Well done !!